# Hedgehog artificial macrophage with atomic-catalytic centers to combat Drug-resistant bacteria

Yanping Long [1], Ling Li [2,3], Tao Xu [1], Xizheng Wu [1], Yun Gao [4], Jianbo Huang [2], Chao He [1], Tian Ma[2], Lang Ma [2], Chong Cheng [1✉] & Changsheng Zhao [1,5,6✉]

Pathogenic drug-resistant bacteria represent a threat to human health, for instance, the methicillin-resistant *Staphylococcus aureus* (MRSA). There is an ever-growing need to develop non-antibiotic strategies to fight bacteria without triggering drug resistance. Here, we design a hedgehog artificial macrophage with atomic-catalytic centers to combat MRSA by mimicking the "capture and killing" process of macrophages. The experimental studies and theoretical calculations reveal that the synthesized materials can efficiently capture and kill MRSA by the hedgehog topography and substantial generation of $\bullet O_2^-$ and HClO with its $Fe_2N_6O$ catalytic centers. The synthesized artificial macrophage exhibits a low minimal inhibition concentration (8 μg/mL Fe-Art M with $H_2O_2$ (100 μM)) to combat MRSA and rapidly promote the healing of bacteria-infected wounds on rabbit skin. We suggest that the application of this hedgehog artificial macrophage with "capture and killing" capability and high ROS-catalytic activity will open up a promising pathway to develop antibacterial materials for bionic and non-antibiotic disinfection strategies.

---

[1] College of Polymer Science and Engineering, State Key Laboratory of Polymer Materials Engineering, Sichuan University, 610065 Chengdu, China. [2] Department of Ultrasound, West China Hospital, Sichuan University, 610041 Chengdu, China. [3] Department of Ultrasound, Affiliated Hospital of North Sichuan Medical College, 637000 Nanchong, China. [4] College of Biomass Science and Engineering, Textile Institute, Sichuan University, 610065 Chengdu, China. [5] College of Biomedical Engineering, National Engineering Research Center for Biomaterials, Sichuan University, 610064 Chengdu, China. [6] College of Chemical Engineering, Sichuan University, 610065 Chengdu, China. ✉email: chong.cheng@scu.edu.cn; zhaochsh70@scu.edu.cn

Accompanied by the overuse of antibiotics, a series of pathogenic bacteria are becoming the so-called drug-resistant bacteria, which puts numerous lives in danger and becomes an ever-increasing crisis[1–5]. As a representative pathogenic drug-resistant bacteria, the methicillin-resistant *Staphylococcus aureus* (MRSA) can resist many types of common antibiotics[6]. Currently, there is an ever-growing need to develop non-antibiotic strategies to fight bacteria without triggering drug resistance, and numerous efforts have been paid to this field with the advancements of nanoparticles (NPs), for instance, loading multiple antibacterial molecules into NPs to reduce the formation of drug-resistance[7], and tuning geometry of NPs to enhance the enzyme inhibition and antibacterial activities[8]. Recently, generating reactive oxygen species (ROS) via chemodynamic, photodynamic, sonodynamic, and enzyme-mimetic materials to combat bacteria has also been explored as promising alternative antibacterial strategies[9–11]. Among diverse ROS generation materials, the enzyme-mimetic catalysts, including nanoliposome[12], metal-organic frameworks[13–15], and inorganic materials[16,17], have gained tremendous popularity by catalyzing oxygen and hydrogen peroxide ($H_2O_2$) to generate ROS via the existing natural pathway. Nevertheless, when treated at a very low concentration, such enzyme-mimetic catalysts present insufficient bacterial killing activities, while utilizing antibacterial materials at high concentrations will always lead to a severe concern on biocompatibility. Thus, developing more effective ROS-based enzyme-mimetic strategies to fight drug-resistant bacteria is of great importance in current material and biomedical sciences.

The human immune system has provided us with an ideal example to overcome this bionic challenge; it has been clarified that macrophages play a critical role in disinfection and wound healing[18–20]. The macrophage can efficiently eradicate pathogenic bacteria by successively capturing, phagocytosing, and killing actions. The macrophage exhibits a rough surface with microfold, prominences, and recognition receptors, which facilitate the detection and capture of bacteria. After phagocytosis, various peroxidases (POD) in the lysosome can produce abundant localized ROS to kill phagocytosed bacteria rapidly[21,22]. Recently molecular biology reveals that these ROS-generation enzymes mainly consist of heme-containing catalytic sites, which can generate highly toxic hypochlorous acid (HClO) and other types of ROS in the presence of $Cl^-$ and $H_2O_2$[23–25]. Though $V_2O_5$ nanorods have been reported to show excellent haloperoxidase (HPO)-mimetic production activity for HClO, the potential cellular toxicity and poor bacterial capture ability of $V_2O_5$ may limit their application in clinical treatment[26]. Thus, it is a crucial challenge to develop ROS catalytic materials that can simultaneously mimic the "capture and killing" process of macrophages to combat bacteria, especially integrated with heme-mimetic catalytic sites.

To address this challenge, we present here the design of a hedgehog artificial macrophage with heme-mimetic atomic-catalytic centers to combat MRSA. The main motivation of this work originates from two aspects: (1) designing physiologically stable, micron-sized, and hedgehog carbonaceous particles to mimic the topography of macrophage for localized bacterial capture, (2) precisely engineering metal-$N_x$-$C_y$ based single-atom catalytic sites to mimic the heme-structure (Fig. 1a); here the Fe-$N_x$-$C_y$ is the primary candidate, while the V-$N_x$-$C_y$ is synthesized for comparison due to the natural widely existence of Fe-HPO (Fig. 1b) and V-HPO (Fig. 1c) for HClO production. The experimental studies and theoretical calculations reveal that the synthesized artificial macrophage can efficiently capture and kill MRSA by the hedgehog topography and localized massive generation of $\bullet O_2^-$ and HClO with its $Fe_2N_6O$ catalytic centers. The synthesized artificial macrophage exhibits a low minimal inhibition concentration (8 μg/mL Fe-Art M with $H_2O_2$ (100 μM)) to combat MRSA and can rapidly promote the healing of bacteria-infected wounds on rabbits skin, which is comparable to the treatment efficiency of vancomycin.

## Results

**Design and analysis of hedgehog artificial macrophage.** To mimic the "capture and killing" function of macrophages both from the surface topography and ROS generation capability, we designed a micron-sized hedgehog particle doped with Fe-$N_x$ or V-$N_x$ based atomic catalytic centers (Fig. 1d). First, a hedgehog metal-organic framework precursor (Supplementary Figs. and 2) was synthesized by a hydrothermal transformation method with urea and Zn-MOF-74 ([$Zn_2(DOBDC)$, DOBDC = 2,5-dihydroxybenzenedicarboxylic acid]. After carbonization and sublimation of Zn, a highly stable micron-sized hedgehog carbonaceous particle can be obtained and served as the pristine artificial macrophage (Art M) (Supplementary Fig. 3)[27,28]; subsequently, the Fe-$N_x$ or V-$N_x$ catalytic centers are synthesized by the in situ coating and carbonization of metal-phenanthroline complex (Supplementary Fig. 4)[29]. Then, the synthesized Art M and Fe-doped or V-doped-Art M (named as Fe-Art M or V-Art M) are applied to act as artificial macrophages to capture and kill MRSA by the hedgehog topography and massive generation of ROS with its POD-/HPO-mimetic catalytic activities (Fig. 1e). Interestingly, it is also noticed that the carbonaceous Fe-Art M exhibits good dispersibility in diverse aqueous media than Fe-doped carbonous micro-particles (Fe-C-MPs, smooth surface) (Supplementary Figs. 5–7). An earlier report has suggested that the hedgehog structure can efficiently enhance the aqueous dispersibility of particles due to the decreased contact area, trapping of air, autoionization of water, etc.[30], which may also facilitate the bacterial capture capability.

The representative hedgehog morphology and the spikes of different types of Art Ms are verified by scanning electron microscopy (SEM) and high-resolution transmission electron microscopy (HR-TEM) (Fig. 2a, b and Supplementary Figs. 8–9). In the meantime, the surface area and pore size distribution of the carbonaceous particle are analyzed by the Brunauer–Emmett–Teller (BET) method. All the Art Ms show high surface areas (up to 1895.8 $m^2/g$, Fig. 2c) with abundant pores ranged from 1.01 to 4.84 nm (Supplementary Fig. 10). The carbon structure and metal phases of the Art Ms are detected by X-ray diffraction (XRD) pattern, and the existing weak characteristic peaks of carbon (002) and (101) diffractions suggest that all of these porous Art Ms display low graphitic degree with abundant defects and no trace of metal-related phases as observed in the HRTEM image (Fig. 2d). This unique porous carbonaceous structure may greatly facilitate the transportation of substrates during the ROS catalytic reactions. Furthermore, the high-angle annular dark-field scanning transmission electron microscopy (HAADF-STEM) images and corresponding energy dispersive spectroscopy (EDS) elemental mapping images show that atomic level Fe and V atoms are homogeneously distributed on the hedgehog particles (Fig. 2e and Supplementary Fig. 11).

To directly demonstrate the concept that these hedgehog artificial macrophages can be applied to capture the bacteria via its spiky topography and bioadhesive carbonaceous structure[32], we have incubated the MRSA with different Art Ms and other comparison samples (Fe-C-MPs, smooth surface, Supplementary Fig. 12, polystyrene microspheres (smooth surface), $SiO_2$ microsphere (slightly rough surface)). It can be observed that all types of Art Ms can efficiently capture abundant bacteria with no significant difference, as demonstrated by the SEM and fluorescence microscopy images (Fig. 2f–j and Supplementary

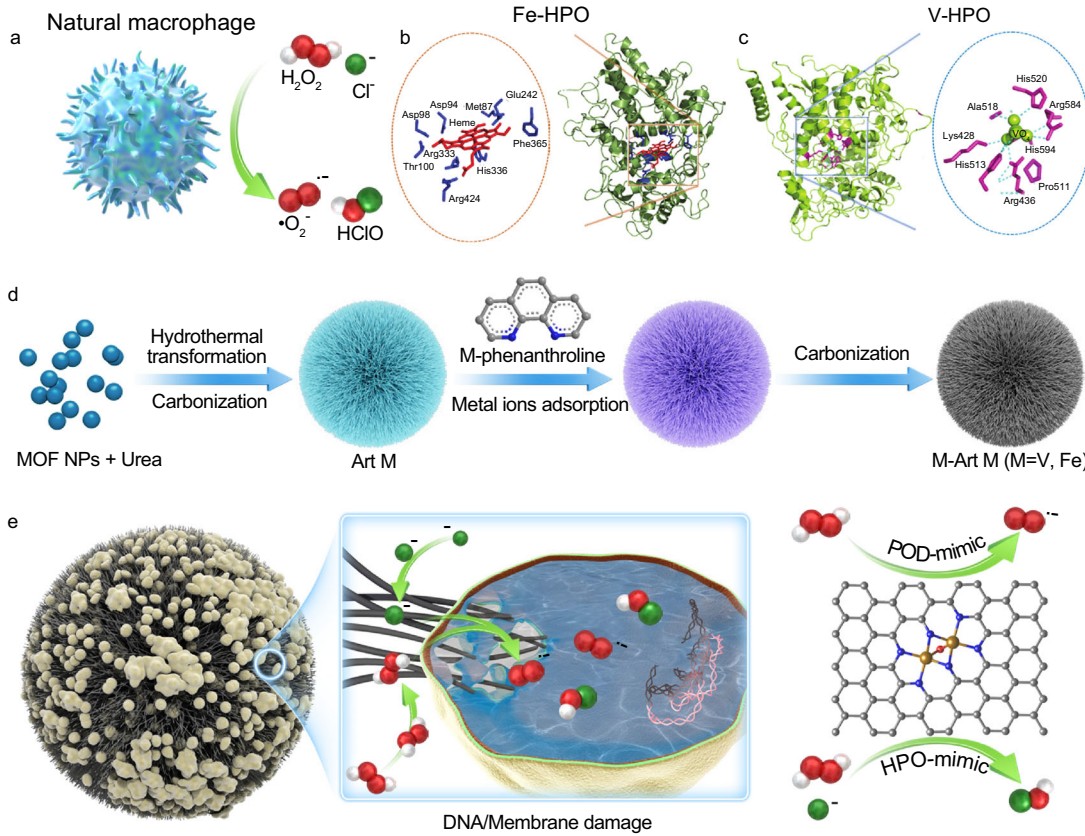

**Fig. 1 Design of hedgehog artificial macrophage. a** Illustrative image to show that natural macrophages can produce ●$O_2^-$ and HClO in the human body. **b**, **c** Scheme of heme-iron-dependent haloperoxidases (Fe-HPO) with Fe as catalytic site and vanadium-dependent haloperoxidases (V-HPO) with V as the catalytic site. These figures were produced with permissions[31]. Copyright 2020, Taylor & Francis Inc. **d** Schematic illustration of the fabrication process of pristine Art M and V/Fe-Art M. **e** Representative working principle of Fe-Art M for localized MRSA eradication by the "capture and killing" function.

Figs. 13–17). The results of Fe-C-MPs and $SiO_2$ microspheres indicate that both the carbonous surface and slightly rough surface exhibit limited bacterial capture capability (Supplementary Figs. 18–21); while the polystyrene microspheres with the smooth surface show nearly no bacterial capture capability (Supplementary Figs. 22 and 23). What's more, for the pristine Art M without doping of Fe or V, there is a very limited amount of dead bacteria, thus indicating that the physical penetration of spikes can't efficiently kill bacteria. For the Fe-Art M, most of the captured MRSA are dead, while the bacteria surround the Fe-Art M show good viability (Supplementary Fig. 16). These data indicate that the Fe-Art M and V-Art M exhibit contact-killing activity to bacteria. As presented in Supplementary Fig. 24, we have analyzed the intrinsic oxidase-mimetic activity of these Art Ms, which suggests that the Fe-Art M exhibits very high catalytic transformation of oxygen into ROS via activation and cleavage of the O–O bond[33]. This good contact killing activity of Fe-Art M preliminarily supports the proposed concept for the bacterial "capture and killing" function of hedgehog artificial macrophage.

**Atomic catalytic centers analysis of Fe-Art M.** As verified that the Fe-Art M exhibits the best contact killing activity to MRSA, in the following section, we will mainly focus on exploring the detailed characteristics of Fe-Art M. First, we use X-ray photoelectron spectroscopy (XPS) to reveal the atomic catalytic centers of Fe-Art M. The N and Fe contents in Fe-Art M are 1.64 and 0.11 at.% with a shift of pyridinic N in Fe-Art M (398.89 eV) when compared to that in Art M (398.49 eV), indicating the formation of $Fe-N_x$/pyridinic-N (Fig. 3a and Supplementary

Tables 2–3). Meanwhile, in Fig. 3b, the Fe 2$p$ spectra can be divided into two peaks belonging to $Fe^{2+}/Fe^{3+}$ ionic species, and no $Fe^0$ and other valence states can be detected[34,35], suggesting the $Fe-N_x$ coordination structure. Besides, the V-Art M (0.10 at.% of V) also reveals similar metal-$N_x$ coordination structures as shown in Supplementary Figs. 25–29.

Then, to reveal the precise atomic coordination $Fe-N_x$ structure, the Fe K-edge X-ray absorption near-edge structure (XANES) was tested for Fe-Art M with Fe foil, $Fe_2O_3$, and Fe-phthalocyanine (FePc) as references (Fig. 3c, d and Supplementary Fig. 30), the pre-edge peak in Fe-Art M positioned at around 7114.6 eV is approximately located between FePc and $Fe_2O_3$, suggesting a different Fe–N structure with higher valency compared to the representative $Fe-N_4$. Thus it is believed that the Fe atoms may carry positive charges and are partially oxidized compared with the $Fe-N_4$ structure in FePc[36]. According to the Fourier-transformed $k^3$-weighted extended X-ray absorption fine structure (FT-EXAFS) spectra, the EXAFS fitting results and corresponding wavelet-transform (WT) images are summarized in Fig. 3e–h and Supplementary Fig. 31 and Supplementary Table 3. The Fe-Art M represents three different bond distances of Fe–O/N (1.85 Å) and Fe–Fe (2.47 Å and 3.18 Å), which are significantly different from the $Fe-N_4$ structure in FePc. Combined with the high coordination numbers of the Fe-N/O (4.9), the existence of Fe-Fe bond, and no $Fe^0$ valence (XPS spectra), it is suggested that a $Fe_2N_{x(x=6-8)}O$ structure exists in the Fe-Art M. Then, we simulate four kinds of possible $Fe_2N_{x(x=6-8)}O$ configurations via the density functional theory (DFT) method and give the corresponding Fe–Fe bond lengths in Supplementary Fig. 32 for further analysis.

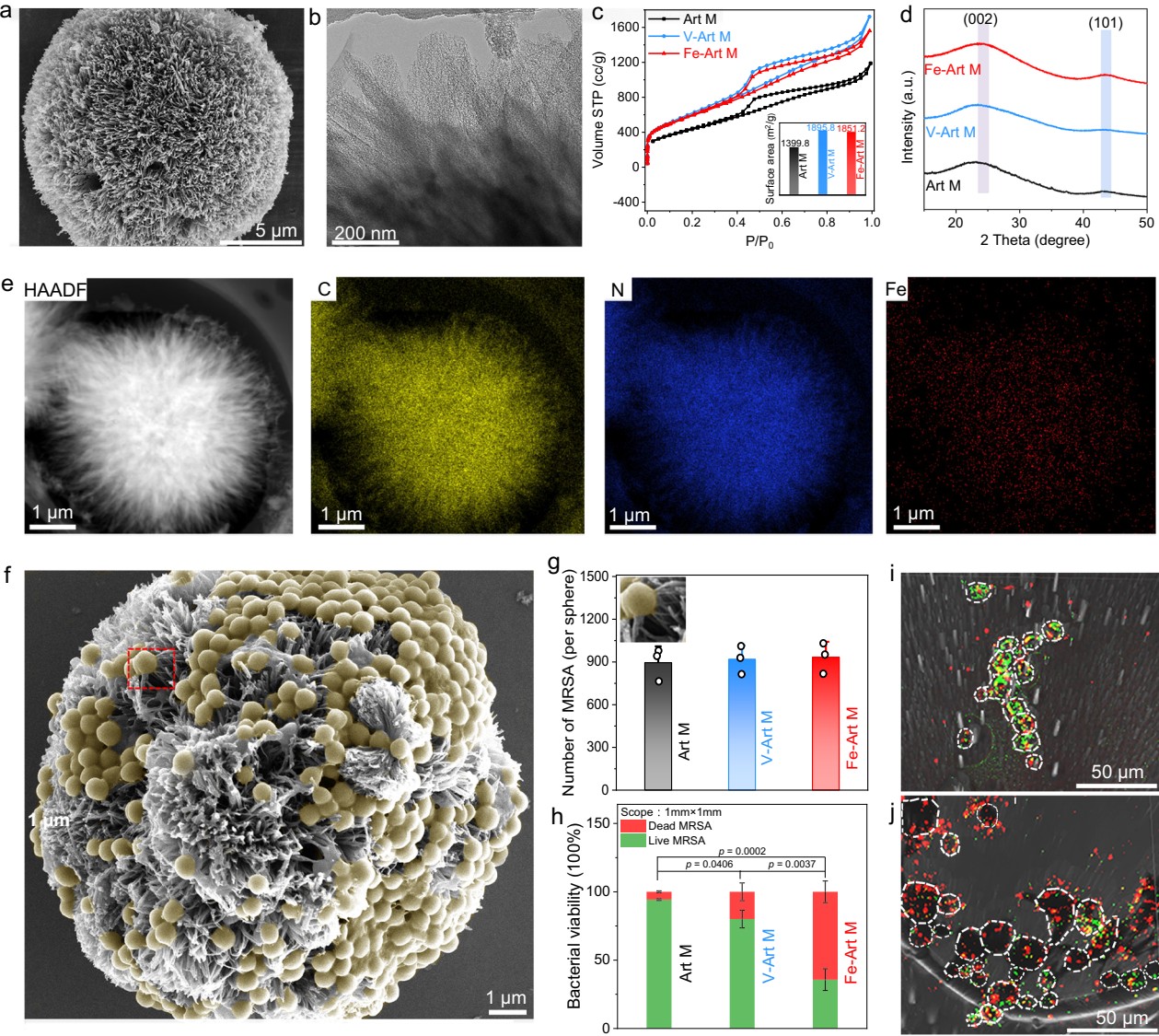

**Fig. 2 Characteristics of different Art Ms and their bacterial "capture and killing" function. a** SEM and **b** HR-TEM images of Fe-Art-M. **c** N$_2$ adsorption/ desorption isotherms and corresponding surface areas of Art M, Fe-Art M, and V-Art M. **d** XRD spectra of Art M, Fe-Art M, and V-Art M. **e** The HAADF-STEM images and corresponding EDS mapping of Fe-Art M. **f** SEM images of the bacterial capture capability of Fe-Art M when incubated with MRSA. **g** The calculated number of MRSA captured by different Art Ms in **e** ($n = 3$ independent experiments, data are presented as mean ± SD). The inset of **g** is a magnified captured bacteria by the spikes. **h** Calculated live/dead bacteria ratio based on the fluorescence results, pH = ~5.8; data were obtained from three fluorescent microscopy images with scope 1 mm × 1 mm, and presented as mean ± SD. 3D reconstructions from confocal laser scanning microscopy (CLSM) images of **i** V-Art M and **j** Fe-Art M when treated with MRSA at the same condition as **h**. **h** $p$ values are assessed by unpaired Student's two-sided $t$-tests. Experiments were repeated independently (**a**, **b**, **e**, **f**, **i**, **j**) three times with similar results. **d** a.u. indicates the arbitrary units. Source data are provided as a Source Data file.

The high-resolution HAADF-STEM images are presented in Fig. 3i, j and Supplementary Fig. 33; there are abundant paired bright atom dots, thus further verifying the Fe$_2$N$_{x(x=6-8)}$O structure[37,38]. The distance shown in the intensity profiles is around 2.20–2.30 Å (Fig. 3k), which is consistent with the major Fe-Fe bond (2.47 Å) detected from EXAFS. Furthermore, the Fe$_2$N$_6$O configuration also fits well with the corresponding Fe K-edge EXAFS for Fe-Art M in R-space (Fig. 3f). Therefore, we conclude that the simulated Fe$_2$N$_6$O configuration with a bond distance of ~2.32 Å (Fig. 3l) matches well with the results of EXAFS (2.47 Å) and high-resolution HAADF-STEM images (2.20-2.30 Å), in which the two Fe atoms are anchored by six N atoms in the graphitic framework, and the Fe–Fe are bridged by one O atom. Thus, Fe$_2$N$_6$O is the major catalytic active center in

the Fe-Art M; meanwhile, the V-Art M also gives similar atomic dispersed structures (Supplementary Fig. 34).

**Evaluation of enzyme-mimetic ROS production activity.** After validation of the enzyme-mimetic catalytic centers of Fe-Art M, we further tested and compared their POD-like and HPO-like activities for ROS production. By using the classic 3,3,5,5-tetra-methylbenzidine (TMB) and o-phenylenediamine (OPD) colorimetric methods, the Fe-Art M shows the best POD-like performance in a pH-dependent manner (Fig. 4a and Supplementary Fig. 35)[39]. Then, we calculate the values of maximal reaction velocity ($V_{max}$), Michaelis constant ($K_m$), and turnover number (TON, the maximum number of conversing substrates of

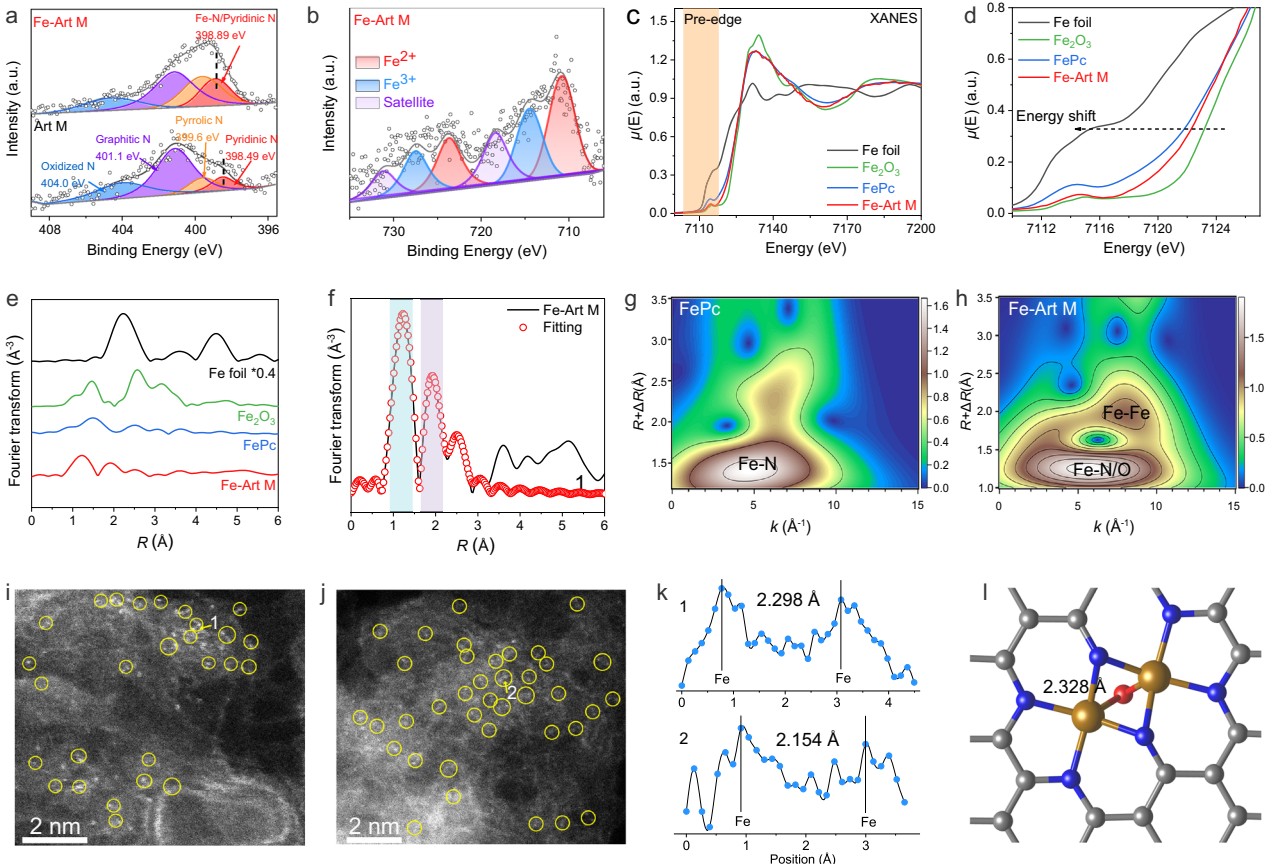

**Fig. 3 Analysis of Fe-N$_x$ catalytic centers for Fe-Art M. a** Comparison of curve-fitted high-resolution XPS N 1s spectra of Art M and Fe-Art M. **b** High-resolution Fe 2p XPS spectra for Fe-Art M. **c** XANES and **d** magnified pre-edge XANES spectra of Fe foil, Fe$_2$O$_3$, FePc, and Fe-Art M. **e** Fourier transformation of Fe K-edge EXAFS of samples in R-space. **f** The corresponding Fe K-edge EXAFS fitting curve for Fe-Art M in R-space. WT-EXAFS of **g** FePc and **h** Fe-Art M. **i, j** High-resolution HAADF-STEM images of Fe-Art M. Experiments were repeated independently **i, j** three times with similar results. **k** Intensity profiles obtained in positions 1 and 2, and the corresponding curves show Fe–Fe projection distances on the visual plane. **l** Simulated precise atomic coordination Fe$_2$N$_6$O structure in Fe-Art M. Atom colors: orange, Fe; gray, C; blue, N; red, O. In **a, b, c, d**, a.u. indicates the arbitrary units. Source data are provided as a Source Data file.

per unit active catalytic center). Compared to the V-Art M ($V_{max} = 5.14$ and TON = 2.73), the Fe-Art M exhibits higher $V_{max} = 11.51$ and TON = 7.49 values and lower $K_m$ ($K_m = 2.53$) (Fig. 4c), thus indicating more efficient catalytic kinetics and affinity of Fe-N active centers toward H$_2$O$_2$. Then, we compare the catalytic activity of the Fe-Art M with the recently reported POD-mimics, including metallic oxides, metal nanoparticles, single-atom enzyme-mimics, and metal-organic frameworks (the literature are shown in Supplementary Table 5). Although not possessing the highest $V_{max}$, the Fe-Art M exhibits the highest TON values among these current established POD-mimics (Fig. 4d). The electron paramagnetic resonance (EPR) and fluorescence analysis reveal that the generated ROS is •O$_2^-$[40], and the Fe-Art M produces a much higher amount of •O$_2^-$ than the V-Art M and pristine Art M (Fig. 4e and Supplementary Fig. 36), which is similar to the natural heme-containing POD[41]. Finally, the production of •O$_2^-$ has also been confirmed by the free radical quenching experiments (Fig. 4f); meanwhile, no obvious generation of •OH has been detected, and the $^1$O$_2$ is originated from pristine Art M particle due to the generation of $^1$O$_2$ did not increase obviously after introducing V/Fe-N structure (Supplementary Figs. 37–39).

The natural HPO can catalyze the halide oxidation by H$_2$O$_2$ to generate OCl$^-$, the catalytic process can be detected by celestine blue (CB) and aminophenyl fluorescein (APF) probe (Supplementary Figs. 40 and 41)[42]. Thus, we utilize the CB as a probe to

measure the HPO-mimetic activity of Fe-Art M[43]. As can be seen in Fig. 4g, h and Supplementary Figs. 41 and 42, the Fe-Art M displays much better catalytic activity than V-Art M and pristine Art M; besides, the Fe-Art M also reveal the best activity under acid condition. Taken together, we have demonstrated that Fe-Art M presents the best POD-like and HPO-like activities for ROS production than V-Art M and pristine Art M, which further proves that Fe-N catalytic centers exhibit the best activity to function as artificial macrophage.

**Theoretical calculation of enzyme-mimetic ROS production pathways.** To elucidate the detailed POD-like and HPO-like ROS catalytic pathways of Fe-Art M, we use the DFT calculations to simulate the •O$_2^-$ and HClO-generating processes. First, we compare the Fe$_2$N$_6$O with Fe$_2$N$_6$O adsorbing H$_2$O$_2$ molecules through the differential charge density, reduced density gradient, and projected density of states (PDOS). At a position of −2.5 eV, the O 2p and H 1s orbitals of H$_2$O$_2$ have a certain weak inter-action with the N 2p orbital, and charge at the OH group transfers to the N direction; besides, the acting force between Fe and O is Van der Waals action (Fig. 5a and Supplementary Figs. 43–45).

Then, we propose three potential POD-like reaction paths and elucidate the Gibbs free energy of all intermediates for each reaction step, resulting in the theoretical free energy for different

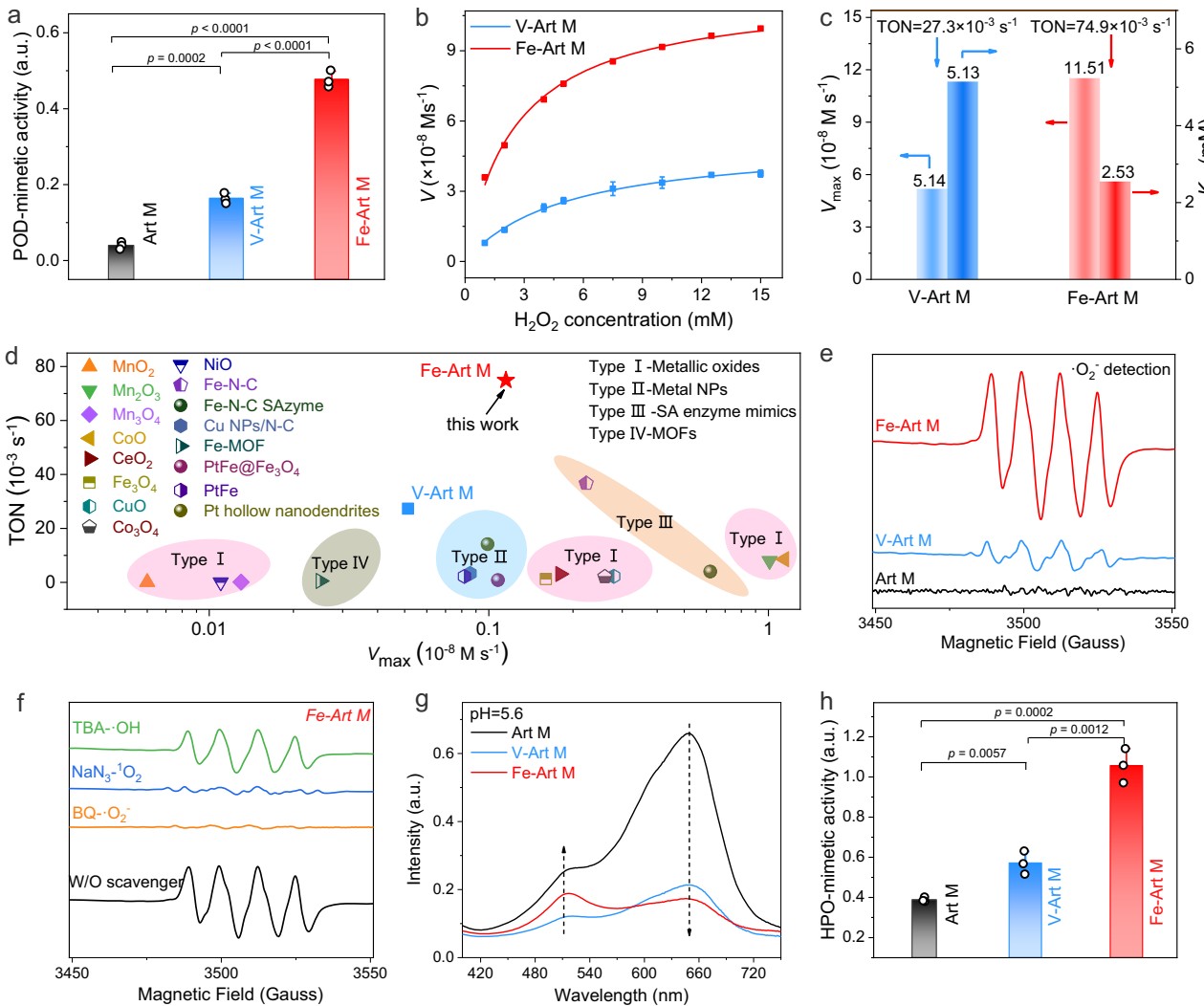

**Fig. 4 Enzyme-mimetic ROS production activity. a** POD-mimetic activity via TMB-based UV–vis spectra in the presence of Art Ms and $H_2O_2$ ($n = 3$ independent experiments, data are presented as mean ± SD). **b** Typical Michaelis-Menten curves. For V-Art $M_{7.5}$, V-Art $M_{10}$, $n = 3$ independent experiments, data are presented as mean ± SD. The rest experiments were repeated three times and show a consistent trend. **c** $V_{max}$, $V_0$, $K_m$, and TON values of V-Art M or Fe-Art M. **d** Comparison of the TON and $V_{max}$ values with reported enzyme-mimetic catalysts. NPs: nanoparticles; MOFs: metal-organic frameworks. **e** DMPO spin-trapping EPR spectra. **f** DMPO spin-trapping EPR spectra of Fe-Art M with different radical scavengers in DMSO solution. Benzoquinone (BQ, •$O_2^-$ quencher), tertiary butanol (TBA, •OH quencher), $NaN_3$ ($^1O_2$ quencher). **g, h** Detection of HPO-mimetic activity using CB as agents, the relative activity is referred to the rate of $Intensity_{512\ nm}/Intensity_{648\ nm}$. (In **h**, $n = 3$ independent experiments per group, data are presented as mean ± SD. In **a, h**, $p$ values are assessed by unpaired Student's two-sided $t$-tests. $V$ is the initial velocity, $V_{max}$ is the maximal reaction velocity, $K_m$ is the Michaelis constant, TON is turnover number. In **a, g, h**, a.u. indicates the arbitrary units. Source data are provided as a Source Data file.

paths (Fig. 5b, c and Supplementary Fig. 46). Figure 5c shows that the $Fe_2N_6O$ in Fe-Art M can efficiently adsorb the $H_2O_2^*$ with a free energy of −0.16 eV. The subsequent cleavage of $H_2O_2^*$ to form two OH* can easily occur with a free energy of −1.05 eV. For reaction path 1, the rate-determining step happens at the third step, the desorption of $H^+$ and formation of O* (iv) exhibit with a relatively low free energy barrier (0.86 eV) compared to the high free energy of OH* desorption (1.51 eV) in reaction path 3, which also theoretically explain why there is no generation of •OH in Supplementary Fig. 37. Then, there are two potential paths, in path 1, an OH* move to O* site and form OOH* (v) with a free energy of 0.71 eV, while in path 2, the formation of two O* will need free energy of 1.70 eV, this should be very difficult to happen compared with path 1. Therefore, we demonstrate that reaction path 1 shows the most reasonable POD-like catalytic processes of Fe-Art.

For the HPO-like catalytic process, we propose two potential reaction paths (Fig. 5d, e and Supplementary Fig. 47). Both paths share the same process of forming two OH*; subsequently, one OH* can easily combine with Cl* species to produce a molecule of HClO with a free energy of −1.67 eV. Then, there are two potential paths; one is to remove the OH* directly with a very high free energy of 2.71 eV, which is supposed to be not reasonable. Another path is to adsorb $H_2O_2$ first and then remove an H* to the OH* to form $H_2O^* + OOH^*$. The OOH* will desorb and become •$O_2^-$ and $H^+$. Therefore, the Fe-Art M could generate •$O_2^-$ and HClO simultaneously in the condition of $H_2O_2$ and $Cl^-$, which agrees well with our experimental results shown in Supplementary Fig. 48.

**Sterilization assessment of drug-resistant bacteria.** After validating that the Fe-Art M can function as an artificial macrophage

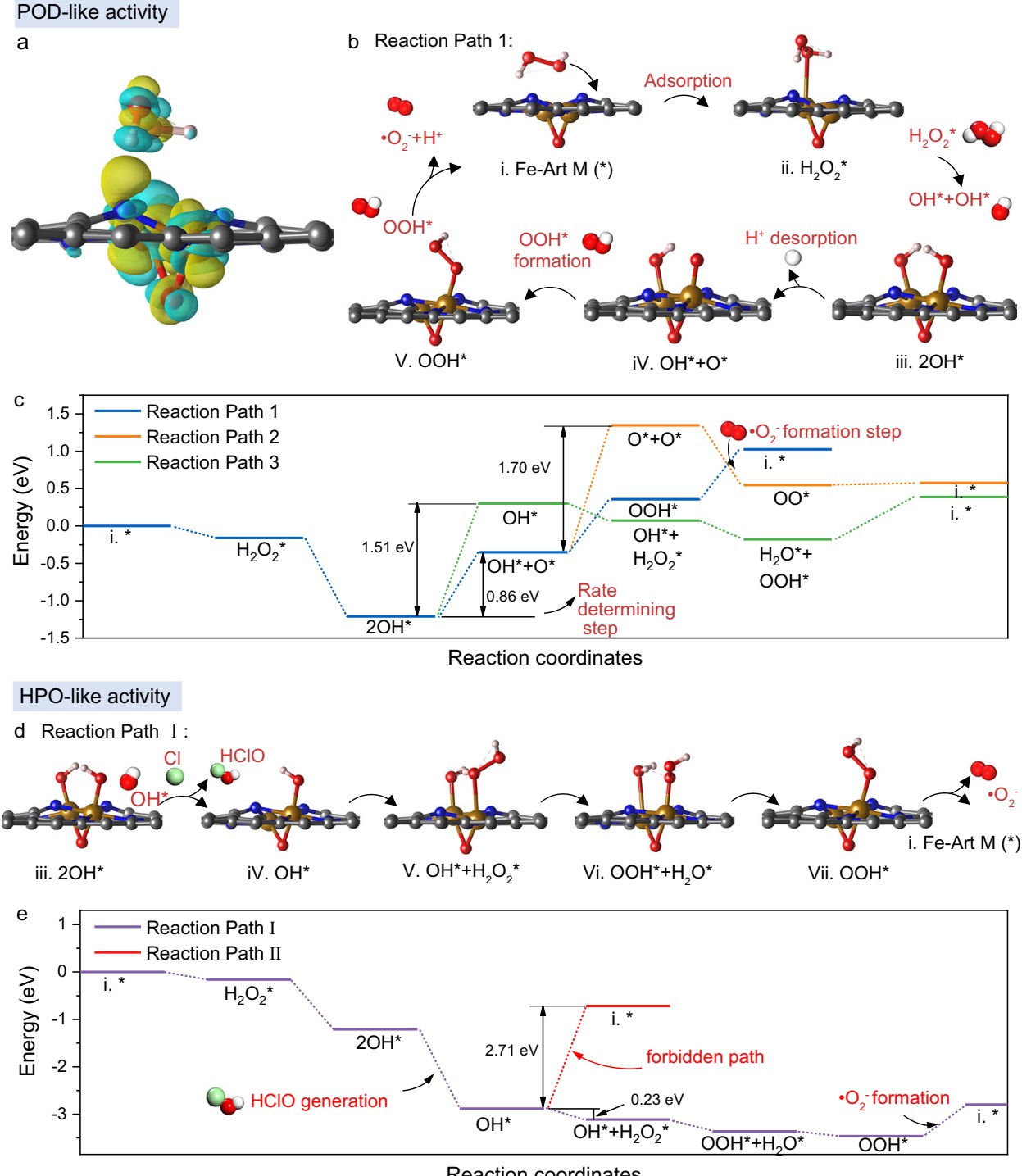

**Fig. 5 Theoretical calculation of POD-like and HPO-like ROS catalytic reaction pathways. a** Electron density difference image of $H_2O_2$ adsorption structure on $Fe_2N_6O$. Yellow contours indicate electron accumulation, and cyan contours denote electron deletion. **b** Proposed reaction path and **c** corresponding free energy diagrams of reaction path 1 and other two potential reaction paths toward generating $\bullet O_2^-$ as POD-mimics. **d** Proposed reaction path and **e** the free energy diagrams toward generating HClO as HPO-mimics. Atom colors: orange, Fe; gray, C; blue, N; red, O; white, H; and green, Cl. Source data are provided as a Source Data file.

to simultaneously capture the bacteria by its hedgehog topography and catalytic generation of ROS with its POD-/HPO-mimetic activities, we then carefully examine its activity in combating drug-resistant bacteria (Fig. 6a). First, we evaluate the cellular compatibility of samples with L929s cells according to the animal wound disinfection process, which shows limited short-term toxicity, and the cells can soon recover after the removal of

materials (Supplementary Figs. 49–52). When incubating the materials with MRSA, the $OD_{600}$ values of the Fe-Art M + $H_2O_2$ group remain no growth of bacteria along with the time period, verifying its remarkable antibacterial activity (Fig. 6b and Supplementary Figs. 53–55). Notably, as displayed in Fig. 6c, the minimal inhibition concentration (MIC) for Fe-Art M + $H_2O_2$ is merely 8 μg/mL Fe-Art M with $H_2O_2$ (100 μM), being close to

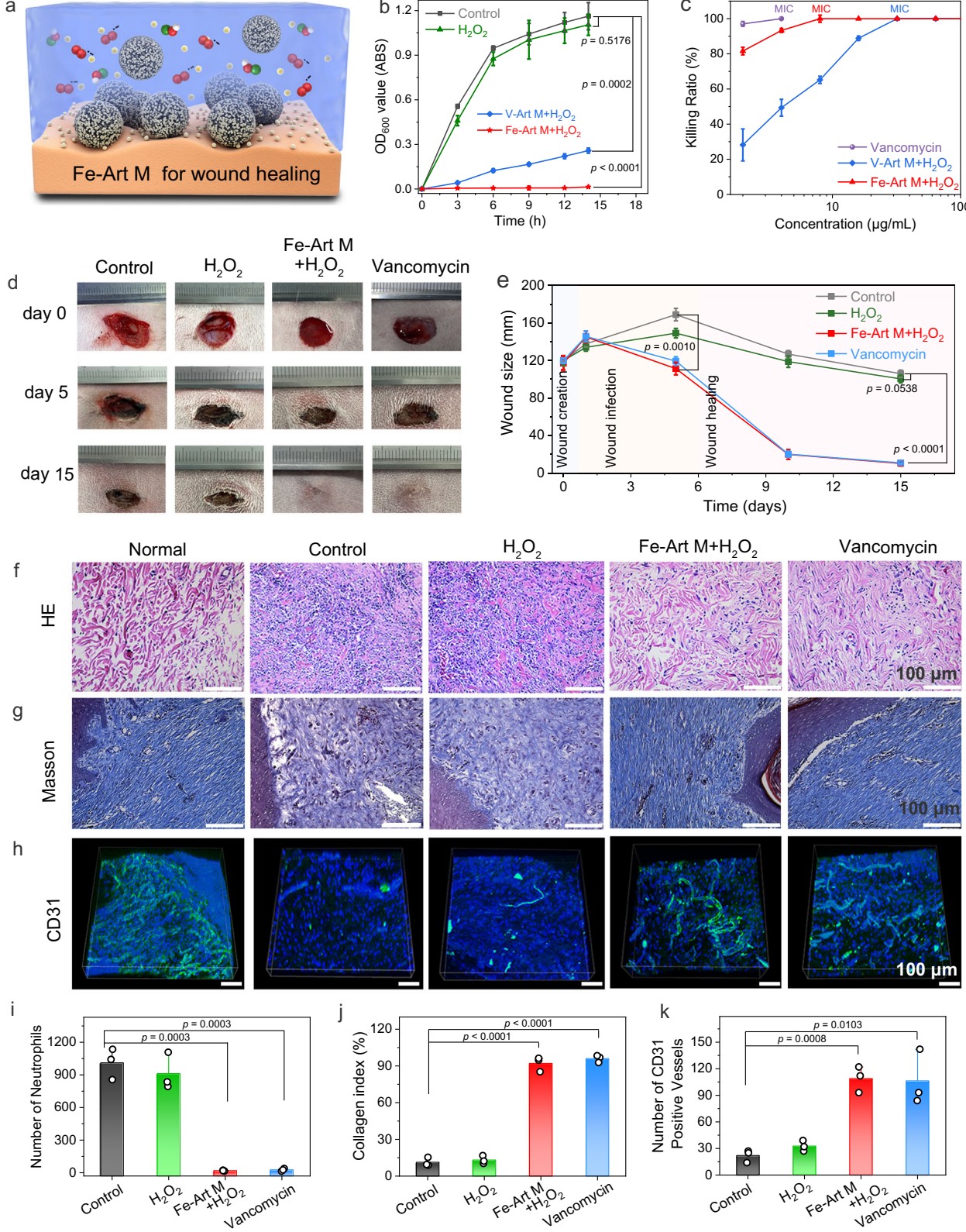

that of vancomycin (4 µg/mL) and much lower than that of 32 µg/mL V-Art M with $H_2O_2$ (100 µM).

Then, to assess the animal wound disinfection activity, MRSA was introduced to infect rabbit skin wounds. The photographs and change of wound sizes were recorded and shown in Fig. 6d, e and Supplementary Figs. 56 and 57. On day 5, the saline and $H_2O_2$ treated groups exhibit severe swelling and pyosis due to the

reproduction of MRSA, while the wounds treated by Fe-Art $M + H_2O_2$ and vancomycin show reduced swelling and wound sizes. After 15 days, the Fe-Art $M + H_2O_2$ and vancomycin groups all healed, whereas the wounds in saline-treated and $H_2O_2$-treated groups still maintain large wound sizes with abscess.

Thereafter, we take the Hematoxylin-eosin (HE), Masson, and CD31 staining to examine the wound tissues and main organs

**Fig. 6 Sterilization assessment of drug-resistant bacteria of Fe-Art M. a** Schematic illustration for the "capture and killing" function of Fe-Art M by the hedgehog topography and massive generation of ROS on the infective wound skin. **b** Real-time $OD_{600}$ values for different samples at 10 μg/mL against MRSA ($n = 3$ independent experiments per group, data are presented as mean ± SD). **c** Killing ratio of MRSA treated with V-Art M and Fe-Art M at different concentrations in the presence of $H_2O_2$ ($n = 3$ independent experiments per group, data are presented as mean ± SD). **d** Digital photographs and **e** the change of wound sizes on different days ($n = 3$ independent experiments per group, data are presented as mean ± SD). **f** HE staining, **g** Masson and **h** CD31 staining images of the epidermal histological sections in different groups at day 15. **i** Number of neutrophils, **j** collagen index, and **k** number of blood vessels. In **i–k**, $n = 3$ independent experiments and data are represented as mean ± SD. In **b, e, i–k**, $p$ values are assessed by unpaired Student's two-sided $t$-tests compared to the control groups. Experiments were repeated independently (**f–h**) three times with similar results. Source data are provided as a Source Data file.

(heart, liver, spleen, lung, and kidney), the data are displayed in Fig. 6f–k and Supplementary Figs. 58–60. The infected wound shows extensive areas of necrotic polymorphonuclear leukocytes and a high number of neutrophils ($1011 ± 116$)[44]; the Fe-Art M + $H_2O_2$ group displays nearly no neutrophils as the vancomycin groups, indicating that MRSA has been eradicated, which is much better than that of the $H_2O_2$ treated group ($912 ± 139$). Besides, no damage or abnormalities are found in the major organs compared to the healthy group, thus suggesting that the skin disinfection treatment of Fe-Art M + $H_2O_2$ is very safe (Supplementary Figs. 58 and 59). Second, as displayed in Fig. 6g, j, for collagen regeneration, we found collagen takes large areas of the tissue slices in the Fe-Art M + $H_2O_2$ ($92 ± 5\%$) and vancomycin ($96 ± 2\%$) groups, and the collagen fibers are uniformly and regularly arranged, thus indicating positive wound-healing effects. Furthermore, as shown in Fig. 6h, k and Supplementary Fig. 60, through the CD31 staining, regenerated endothelial cells in Fe-Art M + $H_2O_2$ ($109 ± 12$) and vancomycin group ($106 ± 25$) can be labeled to reflect newly formed blood vessels after revascularization in the wound area, which is much higher than that of the control ($22 ± 5$) and $H_2O_2$ treated groups ($32 ± 5$). Overall, the above in vivo disinfection results demonstrate that the synthesized Fe-Art M can serve as an antibiotic-free agent to kill MRSA efficiently and promote wound healing rapidly, which has comparable treatment effects as vancomycin.

## Discussion

In summary, our finding demonstrates that the synthesized hedgehog Art M with atomic-catalytic centers is an ideal material to combat MRSA and can mimic the "capture and killing" process of natural macrophage. The electron and fluorescent microscope analyses reveal that the synthesized Art Ms can efficiently capture MRSA by the hedgehog topography. The experimental analyses and theoretical calculations prove that the high catalytic production activity of $\bullet O_2^-$ and HClO are originated from the $Fe_2N_6O$ catalytic centers, and the Fe-Art M exhibits much higher ROS catalytic kinetics than the V-Art M and pristine Art M. The synthesized Fe-Art M exhibits a low minimal inhibition concentration ($8$ μg/mL Fe-Art M with $H_2O_2$ ($100$ μM)) to MRSA and can rapidly promote the wound healing by in vivo disinfection of rabbit skin. We are convinced that this highly ROS-catalytic active Fe-Art M opens up a promising avenue to develop antibacterial materials for bionic and non-antibiotic disinfection strategies.

## Methods

**Materials**. Zinc acetate dihydrate (AR, 99.0%); 2,5-dihydroxyterephthalic acid (98%); urea (≥99.5%); 3,3′,5,5′-tetramethylbenzidine (TMB, standard for GC, ≥99.0%); 5,5-Dimethyl-1-pyrroline N-oxide (DMPO, 97%); vanadium (III) acetylacetonate (99.98% metals basis); iron (III) acetylacetonate (≥99.9% trace metal basis); SLC core–shell monodispersed magnetic silica microspheres (10 mg/mL); polystyrene (melt flow rate, 2.5% w/v); and sodium formate (≥99.5%) were purchased from Aladdin. Ltd. 1, 10-phenanthroline (AR, 99.9%) were purchased from Jilin Chinese Academy of Sciences-Yanshen Technology Co. Ltd. The rest regents, if not sterilized specifically, were supplied by Aladdin. Ltd.

**Synthesis of M-Art M (M = V, Fe)**. The synthetic process of M-Art M was divided into two steps. Firstly, 50 mL of methanol containing zinc acetate dehydrate (0.22 g, 1 mmol) was mixed with 20 mL of methanol containing 2,5-dihydroxyterephthalic acid (60 mg, 0.3 mmol), and the mixture was further ultrasonicated for about 30 min. After washing with water three times, urea (25/100 mg, 0.42/1.68 mmol) in 75 mL deionized water was added in a 175 °C-Teflon-lined autoclave for 24 h. The precipitate was then washed with water and ethanol and dried under vacuum. Afterward, with a heating ramp rate of 5 °C/min, the as-prepared material was transferred into a ceramic boat and placed in a tube furnace, and then heated to 1100 °C for 1 h. Then the products was gathered at room temperature and washed with HCl (1 M) solution at 80 °C for further use. Subsequently, 0.02 g Art M was dispersed in 2.5 mL of ethanol solution, and then a 30 μL mixture (metal acetylacetonate and 1,10 phenanthroline with a molar ratio of 2:1) was added. Finally, the dispersion was dried overnight and heated to 600 °C in an $N_2$ atmosphere for 2 h.

**Characterization of V-Art M and Fe-Art M**. $N_2$ adsorption analysis was conducted on a Quantachrome Autosorb IQ instrument. All powder samples were degassed at 150 °C overnight prior to actual measurement. The surface area was calculated by using BET calculations. The pore size distribution (PSD) plot was recorded from the adsorption branch of the isotherm based on the QSDFT model for slit/cylinder pores. XPS were obtained by using an XPS device (ESCAL 250) to detect the compositions of the M-Art M and analyzed by Avantage to confirm the successful introduction of the metals. SEM images were obtained by using an Apreo S HiVoc (Thermo Fisher Scientific, FEI). Cryo-SEM images were obtained by using Quanta 450FEG (FEI Ltd., USA). TEM images and EDS mapping were obtained via a Talos F200x TEM microscope (FEI Ltd., USA) operated at 200 kV and analyzed by GMS-freeanalysis. XRD pattern presented the crystal phase state via a Bruker D8 Focus X-ray diffractometer and analyzed by MDI Jade and Origin. The X-ray absorption spectra were collected on the beamline BL07A1 in NSRRC and the radiation was monochromatized by a Si (111) double-crystal monochromator. XANES and EXAFS data reduction and analysis were handled via Athena software. Cs-corrected (S)TEM (FEI Titan Cubed Themis G2 300) was used for high-resolution HAADF-STEM.

**Catalytic performance of Fe-Art M**. Peroxidase activities was determined by colorimetric assays. 500 μL of catalysts (0.1 mg mL$^{-1}$), 20 μL of TMB (10 mg mL$^{-1}$) and 25 μL of $H_2O_2$ (1 M) were added into a 2 mL sodium acetate–acetic acid (NaOAc/HOAc) buffer [100 mM (pH 4.5 or 5.6)]. The catalytic oxidation of TMB (oxTMB) was studied by measuring the absorption changes of the oxidized form of TMB at $\lambda_{max} = 652$ nm ($\mathcal{E} = 39,000$ M$^{-1}$ cm$^{-1}$). Unless otherwise stated, POD activities were carried out in an air-saturated buffer. $K_m$ and $V_{max}$ were calculated using Lineweaver-Burk plots of the double reciprocal of the Michaelis–Menten equation $V = V_{max} \times [S]/(K_m + [S])$, $TON = V_{max}/[E]$, where $[S]$ is the concentration of $H_2O_2$, $[E]$ is the molar concentration of metal in nanozymes.

Oxidase-like performance of Fe-Art M. As for the oxidase-like activity test, apart from being the absence of $H_2O_2$, the detection procedure of oxidase performance is similar to that of peroxidase.

Furthermore, the haloperoxidase-mimetic activity test was determined by CB. Five hundred microliter of catalysts (0.1 mg mL$^{-1}$), 5 mL of CB solution (200 μM), 10 μL of $H_2O_2$ (1 M), and 500 μL of NaCl (0.059 g mL$^{-1}$) were added into a 2 mL NaOAc/HOAc buffer [100 mM (pH 5.6 or 7.4)]. The catalytic ability of materials was studied by measuring the absorption changes of CB.

**EPR measurement**. The generation of $\bullet O_2^-$ was evaluated by EPR spectrometer using DMPO spin-trapping adduct in DMSO solvent. Five hundred microliter of catalysts (0.1 mg mL$^{-1}$, DMSO) and 25 μL of $H_2O_2$ (1 M) were was added into a 2 mL DMSO, and then, 10 μL DMPO was added. EPR measurements were performed via the Bruker EPR EMX Plus (Bruker Beijing Science and Technology Ltd, USA) at a frequency of 9.8 GHz (microwave power: 1 mW).

The radical quenching experiment was detected using benzoquinone (1 mg, $\bullet O_2^-$ quencher), NaN$_3$ (1 mg, $^1O_2$ quencher), and tert-Butanol (0.239 mL, $\bullet$OH quencher); the concentrations of each sample are consistent with mentioned above.

The •OH generation activity was detected by DMPO. Five hundred microliter of catalysts (0.1 mg mL$^{-1}$, buffer) and 25 μL of H$_2$O$_2$ (1 M) were added into a 2 mL buffer, and then, 10 μL DMPO was added.

The ability to generate $^1$O$_2$ was detected by 2,2,6,6-Tetramethylpiperidine 1-oxyl (TEMP). 500 μL of catalysts (0.1 mg mL$^{-1}$, buffer) and 25 μL of H$_2$O$_2$ (1 M) were added into a 2 mL buffer, and then, 10 μL TEMP was added.

**Antibacterial experiments**. In vitro. MRSA (ATCC 43300, Gram-positive) and extended-spectrum β-lactamase-producing *Escherichia coli* (ATCC 53104, Gram-negative) were used as the representative pathogenic bacteria to investigate the bacterial capture and eradication abilities of the enzyme-mimetic catalytic systems, the drug-resistance of bacteria is induced by standard protocols. The V-A M and Fe-Art M systems with H$_2$O$_2$ containing NaCl were cultured with 2 mL of bacterial suspensions (10$^7$ CFU mL$^{-1}$) for 14 h at 37 °C. The final concentrations of materials and H$_2$O$_2$ are 10 μg mL$^{-1}$ and 100 μM, respectively. The OD$_{600}$ values of the treated suspensions were measured by a UV–vis spectroscopy (UV-3600, Shimadzu) to investigate the growth of bacteria at 3, 6, 9, 12, and 14 h. Meanwhile, the cultured suspensions were diluted 10$^4$ times and taken for agar plate counting. The antibacterial properties were estimated by comparing the CFUs of different systems with that of the control. Then, OD$_{600}$ values of the bacterial suspensions treated by the systems at different concentrations (4, 8, 16, and 32 μg mL$^{-1}$) were recorded to calculate the MIC values.

The bacterial capture ability and bactericidal performance of these systems were visualized via SEM, cryo-SEM, fluorescence microscopy (DMIRE2, Leica), and CLSM images (N-STORM & A1, Nikon, equipped with NIS-Elements Viewer). After being treated with different samples for 30 min under continuous shaking, the bacterial suspensions were then fixed with 2.5 wt% glutaraldehyde and dehydrated with a gradient of ethanol/water solution. Then, the SEM images were obtained to observe the bacterial capture ability and their morphologies. The in situ cryo-SEM images were also taken directly without fixation, dehydration, and centrifugation steps. Moreover, the bacteria were also stained by LIVE/DEAD BacLight Viability Kits (SYTO-9 for live cells and propidium iodide for dead cells) for observation using fluorescence microscopy and CLSM.

**Cytocompatibility experiments**. For living/dead fluorescence staining, we first co-culture cells with 10 μg/mL particles and H$_2$O$_2$ (100 μM) for 30 min, then change the medium and wash with PBS twice to remove the particles, and then observe live/dead cells immediately or after 1 day. Subsequently, Calcein-AM and PI working solution were added directly to stain cells for 5 min at 37 °C, and washed with PBS. Finally, the cells were immediately visualized by fluorescence microscopy.

For the viability of the L929 (ATCC, CCL-1), we first co-culture cells with 10 μg/mL particles and H$_2$O$_2$ (100 μM) for 30 min, then change the medium and wash with PBS twice to remove the particles, and then observe cell viability immediately or after 1 day. 200 μL of CCK-8-based diluted solution was added to each well, respectively. After 1 h of incubation, the mixture was detected by a microplate reader (Model 550, Bio-Rad) at 450 nm and analyzed by a standard protocol to calculate the cell viability.

**Wound disinfection experiments**. In vivo: Rabbits (New Zealand White Rabbit, male, 8–12 weeks) were selected as models in animal experiments, and all the studies on these animals were performed by following the animal ethical standard from Animal Ethics Committee in West China Hospital, Sichuan University, Chengdu, China. The protocols of all the animal experiments were permitted and carried out as requested. The infected wound model was made by two main steps: i) removal of a hole around 1 cm in diameter on the epidermis; ii) introduction of 100 μL of MRSA suspension (1 × 10$^8$ CFU mL$^{-1}$) and incubation for 1 day. After that, Fe-Art M, H$_2$O$_2$, and diluted saline were added to treat the infected wound. The final concentrations of materials and H$_2$O$_2$ are 10 μg mL$^{-1}$ and 100 μM, respectively. Normal saline, H$_2$O$_2$ (100 μM), and vancomycin (16 μg mL$^{-1}$) were also used as a contrast. The antibacterial treatment time is 30 min; after the disinfection process, the PBS solution was used to wash and remove the residual samples. To observe the residual bacteria in the wound after disinfection treatment, 5 μL of fluids were collected from different wounds, respectively, for agar plate counting. After 16 days, the treated wounds were cut off and fixed with 10% formaldehyde solution for histological HE, Masson, and CD31 staining analysis. The HE and Masson images were taken by optical microscope. The CD31 immunofluorescence images were taken by a CLSM.

**DFT calculations methods**. The DFT calculations were performed by Vienna Ab initio Simulation Package (VASP)[45] with the projector augmented wave (PAW) method[46]. The exchange-functional was treated with the generalized gradient approximation (GGA) of Perdew–Burke–Ernzerhof (PBE)[45] functional. The plane-wave basis's cut-off energy is set at 450 eV for optimizing cell and atoms calculations. The vacuum spacing being perpendicular to the plane of the catalyst is at least 15 Å. The Brillouin zone integration is performed on the primitive cell using 3 × 3 × 1 Monkhorst-Pack k-point[47]. The onvergence energy threshold of 10$^{-5}$ eV is applied for self-consistent calculations. The equilibrium lattice constants within 0.03 eV/Å are optimized for the maximum stress on each atom. As a key element in

the DFT + U method, it is widely believed that the localized 3d electrons correlation of transition metal in the fourth period can be described by considering on-site coulomb (U) and exchange (J) interactions. Furthermore, we applied DFT + U through the rotationally invariant approach with the corresponding U – J values being 3.29[48].

Gibbs free energy is acquired via adding corrections including entopic (TS) and zero-point energy (ZPE) to calculated DFT energy, hence $\Delta_G = \Delta E_{DFT} + \Delta_{ZPE} - T\Delta_S$, in which $\Delta E_{DFT}$ means calculated DFT reaction energy, $\Delta_{ZPE}$ means changes in ZPE calculated from the vibrational frequencie, $\Delta_S$ means changes in the entropy regarding thermodynamics databases.

**Statistical analysis**. All data were expressed in this manuscript as mean ± SD. All the results have been performed three times by independent experiments. A two-tailed Student's *t*-tests was used to analyze the statistical significance between the two groups by using GraphPad Prism 7.0 (GraphPad Software Inc.).

**Reporting summary**. Further information on research design is available in the Nature Research Reporting Summary linked to this article.

## Data availability

All data are available in the main text or the Supplementary Information. Source data are provided with this paper.

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

## Acknowledgements
This work was financially supported by the National Key R&D Program of China (2019YFA0110600, 2019YFA0110601, and 2016YFC1103000), National Natural Science Foundation of China (Nos. 52173133, 82102064, 82102065, 82071938, 82001824, 82001829, and 51803134), the Science and Technology Project of Sichuan Province (Nos. 2021YFH0087, 2021YFH0135, 2021YFS0050, 2021YJ0434, 21YYJC2714, 21ZDYF3763, 2021YFH0180, 2020YFH0087, and 2020YJ0055), China Postdoctoral Science Foundation (2021M692291, 2021M692288), and the 1·3·5 Project for Disciplines of Excellence, West China Hospital, Sichuan University (No. ZYJC21047). Prof. Cheng acknowledges the financial support of the State Key Laboratory of Polymer Materials Engineering (Grant No. sklpme2021-4-02), Fundamental Research Funds for the Central Universities, and Thousand Youth Talents Plan. We thank the Ceshigo for the technical supports on theoretical calculations and XAS. We also thank Zhongkebaice Technology Service Co. Ltd. for the technical supports high-resolution HAADF-STEM and XPS. We gratefully acknowledge Dr. Sujiao Cao, Dr. Mi Zhou, Dr. Minghua Zhang, Dr Xi Xiang, Dr. Yuanjiao Tang, Ms. Xijing Yang, and Ms. Zhen Yang at Sichuan University for the discussion on animal tests. We also thank Li Li, Fei Chen, and Chunjuan Bao of the Institute of Clinical Pathology, Sichuan University, for dealing with histological staining.

## Author contributions
Y.P.L., C.C. and C.S.Z. proposed the idea and designed the experiments. Y.P.L., L.L., T.X, Z.X.W., C.H. and Y.G performed the experiments, characterization and results analysis. J.B.H. and T.M. assisted with the figure production and experiment design. Y.P.L., C.C., C.S.Z. and L.M. design and conduct the theoretical calculation. Y.P.L., C.S.Z. and C.C. wrote and edited the manuscript. C.C. and C.S.Z. supervised the whole project. All authors discussed the results and commented on the manuscript.

## Competing interests
The authors declare no competing interests.

## Additional information

**Peer review information** *Nature Communications* thanks Lizeng Gao and the other anonymous reviewer(s) for their contribution to the peer review this work. Peer reviewer reports are available.

