## [Peer Review File · Nature Communications]

Reviewers' Comments:

Reviewer #1:

Remarks to the Author:

Yanping Long et al address the problem of antibiotic resistant bacteria and in particular the suppression of methicillin-resistant staphylococcus aureus (MRSA). This is a tremendously widespread health issue that remains poorly addressed. In fact, pharmaceutical companies are leaving it by wayside because there are too few novel research leads in this area.

The authors describe a novel approach to it using biomimetic hedgehog particles serving as artificial macrophage. The catalytic activity of these particles based on Fe₂N₆O catalytic centers mimics the chemistry of actual macrophages producing a large amount of •O₂ radicals. The authors found that these artificial macrophage inhibit the MRSA at the concentration as low as 8 µg/mL, which is indeed efficient. Analysis of different POD mimics in Fig 4 is quite useful.

Overall, I believe the manuscript is suitable for the publication in NComm after revisions.

1. I think the authors used the right approach to kill bacteria replicating the biological mechanisms of macrophages. The paper would benefit from the comparison to other NP-based methods of MRSA suppression.

<https://pubs.acs.org/doi/abs/10.1021/acsnano.5b03247>

<https://www.frontiersin.org/articles/10.3389/fphar.2019.01121/full>

The reason for this request is to highlight the differences in the mechanism of actions, because they can be quite different: ROS vs metabolic inhibition. Both of them are protein mimetic

2. The strategy capture and kill shows its efficiency. It seems to me a part of the reason that the Art M are not agglomerating in dispersion. This is the direct consequence of their enhanced dispersability.

3. DFT calculations typically have hard time describing the effect of water media. How did the authors approached this problem here?

Reviewer #2:

Remarks to the Author:

The authors have prepared a hedgehog-like micrometer-sized particle and evaluated the potential of this particle as disinfectant for promoting wound healing. I don't think this work is suitable for publication in Nature Communications due to the following major concerns and inconsistency and would recommend submission somewhere else like Chemical Communications.

1, The scientific question or technical limitation which this work is to address is not clear. On page 3 in the introduction section, the authors stated that "...when treated at a very low concentration, such enzyme-mimetic catalysts present insufficient bacterial killing activities, while utilizing antibacterial materials at high concentrations will always lead to a severe concern on biocompatibility...". But, unfortunately, the hedgehog-like particle reported in this work is cytotoxic to mammalian cells as well, according to Supplementary Figure 32, which though a qualitative assay revealed dead cells (in red) to appreciable ratio no matter whether or not external H₂O₂ was added. In fact, the authors need to quantitatively and qualitatively characterize the antibacterial activity and cytotoxicity of their particle. As summarized by the authors in Supplementary Table 4, there are already quite many reports on nanozymes and I believe that there more nanozymes in the literature than what had been summarized here in Supplementary Table 4. Compared with the previously reported nanozymes, what advantages or uniqueness does the hedgehog particle reported in this manuscript have? If it is just another particle that has catalytic activity, generates ROS, and kills bacteria as do so many of its prior counterparts have already done, then its novelty is limited and it should not be acceptable for publication in Nature Communications.

2, The authors stated that their hedgehog-like particle can capture the bacteria and then kill the as-captured bacteria on contact. A question which the authors failed to address with experimental

evidences is why their particle can capture the bacteria. They claim that their particle has rough surface and, as a result, should be able to capture bacteria nearby. Such a claim is not persuasive and needs support from experimental evidences. I would suggest the authors to prepare particles of similar size and surface chemistry as the hedgehog-like one here but instead with smooth surface and check whether bacterial cells adsorb onto this smooth-surfaced counterpart.

3, Throughout the manuscript, the authors mentioned three different types of enzyme-mimicking activities their particle exhibited, which are oxidase-like on page 6, peroxidase (POD)-like on page 9, and haloperoxidase (HPO)-like also on page 9 (and of course later on other pages). Oxidase, peroxidase, and haloperoxidase use different substrates and correspond to different ROS-generating reaction routes. On page 6, when the authors claimed their particle to be able to convert O₂ into ROS, they used Supplementary Figure 11 to support this claim. Nevertheless, Supplementary Figure 11 did not convey any information on whether O₂ is a substrate for the ROS generation mediated by this hedgehog-like particle; instead, it only shows that there is ROS production at pH 4.5, a very acidic pH. In fact, in both the in vitro antibacterial assays and the rabbit model study, the authors added H₂O₂ (10 mM in the former while 100 mM in the latter) externally in addition to their particle. If their particle is oxidase-like, why should the authors spend extra efforts to add external H₂O₂? If externally added H₂O₂ is necessary, then the potential administration of their hedgehog particle is superficial sites like wounded skin; otherwise, where can such high H₂O₂ dose come from naturally or endogenously? According to the animal study in this work, the hedgehog-like particle did not offer better therapeutic efficacy than vancomycin. Then why should the audience care about using such an inorganic particle to do a job which an available antibiotic can well accomplish?

4, The authors claimed that their hedgehog-like particle produce •O₂ and HClO as the ROS species based on their calculations. They need do experiments to examine whether their particle does produce these ROS species and whether there are other ROS species it produces.

5, On page 9, the authors claimed that their "Fe-Art M shows the best POD-like performance in a pH-independent manner (Fig. 4a and Supplementary Fig. 20)". To support this claim, the authors need to use a pH-independent ROS probe, rather than TMB which works only at acidic pH, and monitor the ROS production under at least three different pH conditions, better to span the acidic, neutral, and basic ranges.

6, On page 14, the authors claimed that their hedgehog-like particle "...shows limited toxicity, especially in the absence of H₂O₂, ... (Supplementary Fig. 32)". But, in Supplementary Fig. 32, both Fe-Art M and V-Art M caused certain ratio of cells to be stained in red (i.e., dead cells), not even to mention those that were deactivated and washed away in the sample preparation. In addition, Supplementary Fig. 32 reveals no detectable difference to the naked eyes between groups in the presence and absence of H₂O₂. In fact, to assess whether their particles are cytotoxic or not, the authors should have done cytotoxicity assays to obtain quantitative evaluation in addition to the qualitative examinations as reported in Supplementary Fig. 32.

7, The authors need to quantitatively check the internalization efficiency of their micrometer-sized particles by mammalian cells.

8. The authors need to examine whether their micrometer-sized particles aggregate under protein-present conditions, to simulate the environment their particles encounter in a wound, their designed administration site.

Reviewer #3:

Remarks to the Author:

In this paper, the authors synthesized a hedgehog Art M with atomic-catalytic centers to combat MRSA by mimicking the "capture and killing" process of natural macrophage integrating nanozyme-catalyzed bacterial killing via POD-like and HPO-like activities. The structure of catalytic site and catalytic mechanism are well analyzed. Such strategy is very interesting to develop artificial system for anti-infection treatment. I think that it merits the consideration for publication on Nature Communications. However, the below questions and concerns need to be addressed before acceptance.

1. The authors mentioned that hedgehog artificial macrophages based on Fe-Art M can capture bacteria via its spiky topography and bioadhesive carbonaceous structure. What is the interaction between bacteria and hedgehog structure to achieve strong "capture"? Is the capture specific for

MRSA or broad for any bacteria, fungi or cells?

2. In Figure 2h, i, j, the authors described that "For the Fe-Art M, most of the captured MRSA are dead, while the bacteria surround the Fe-Art M all show good viability". It is hard to understand such difference. How does Fe-Art M kill bacteria if no hydrogen peroxide is present.

3. For the structure of active site in Fe-Art M, the authors concluded that the Fe-Art M have two bond distances of Fe-O/N (1.85 Å). But the evidence for Fe-O bond is not well provided.

4. What is the optimal pH for POD-like activity and HPO-like activity, respectively? How to coordinate the two activities for bacterial killing.

5. For antibacterial test, 10 mM H₂O₂ is chosen in in vitro and in vivo models. How to determine 10 mM is the best concentration for these experiments. Otherwise, 10 mM is a pretty high concentration which may effectively kill bacteria or mammalian cells.

6. Based on the description in Fig. 6c, the authors concluded that the minimal inhibition concentration (MIC) for Fe-Art M+H₂O₂ is merely 8 µg/mL. Does such MIC refer to the concentration of Fe-Art M? However, as mentioned in the Figure 2 by the authors, only Fe-Art M showed no significant antibacterial activity. Thus, it is questionable to use MIC to determine the antibacterial performance for Fe-Art M.

7. Considering the biocompatibility, can Fe-Art M be degraded under physiological condition?

Thank you for providing the reviewers' comments for our article entitled "*Hedgehog Artificial Macrophage with Atomic-Catalytic Centers to Combat Drug-Resistant Bacteria*" with the manuscript number: NCOMMS-21-12581A.

First, we would like to thank you for giving the opportunity to revise the manuscript. We also want to thank all the reviewers for their valuable scientific comments. In the revised manuscript, with all the useful comments from the reviewers, we have made significant revisions and added all necessary experiments as requested by the reviewers to make the contents and conclusion more accurate; for instance, the capture experiments of different comparison samples toward MRSA and the extended-spectrum β -lactamase-producing *E. coli*, and the corresponding cellular compatibility have been carefully evaluated. Meanwhile, all the questions and concerns from the reviewers have been well-addressed and explained in our revised manuscript and also in this response letter. All the corrections and changes in the revised manuscript are highlighted in red. What's more, the files of "Reporting summary" and "Editorial policy checklist" have also been provided. All the histograms have been revised to bar charts based on your suggestion.

Through the introduced significant changes to the manuscript, we hope that we have merited the revised manuscript for publication in *Nature Communications*. Please find below a list of our detailed responses to all the comments and corrections in the revised manuscript.

Point-by-point response to the detailed comments by reviewers of "*Hedgehog Artificial Macrophage with Atomic-Catalytic Centers to Combat Drug-Resistant Bacteria*" with manuscript ID: NCOMMS-21-12581A.

REVIEWER COMMENTS

Reviewer #1:

“Yanping Long et al. address the problem of antibiotic resistant bacteria and in particular the suppression of methicillin-resistant staphylococcus aureus (MRSA). This is a tremendously widespread health issue that remains poorly addressed. In fact, pharmaceutical companies are leaving it by wayside because there are too few novel research leads in this area.

The authors describe a novel approach to it using biomimetic hedgehog particles serving as artificial macrophage. The catalytic activity of these particles based on Fe_2N_6O catalytic centers mimics the chemistry of actual macrophages producing a large amount of $\bullet O_2^-$ radicals. The authors found that these artificial macrophage inhibit the MRSA at the concentration as low as $8 \mu g/mL$, which is indeed efficient. Analysis of different POD mimics in Fig 4 is quite useful.

Overall, I believe the manuscript is suitable for the publication in NComm after revisions.”

Response to the general comment:

Thanks a lot for your helpful comments and suggestions to further improve the quality of this manuscript. Now the manuscript has been thoroughly revised based on your comments and the other reviewers' comments; we believe that the quality of this paper has been significantly enhanced. We thank you again for your great efforts.

(1) *“I think the authors used the right approach to kill bacteria replicating the biological mechanisms of macrophages. The paper would benefit from the comparison to other NP-based methods of MRSA suppression.*

<https://pubs.acs.org/doi/abs/10.1021/acsnano.5b03247>

<https://www.frontiersin.org/articles/10.3389/fphar.2019.01121/full>

The reason for this request is to highlight the differences in the mechanism of actions, because they can be quite different: ROS vs metabolic inhibition. Both of them are protein mimetic.”

Response to comment:

We thank the reviewer for the valuable suggestions and positive comments on our MRSA killing approach by replicating the biological mechanisms of macrophages. We agree that the comparison of this

study with the other nanoparticles-based methods to kill bacteria and MRSA is highly useful for further understanding the antibacterial mechanisms and validating the novelty of this study.

The continuous exploration and expansion of new nanotechnology have opened up a wide range of potential applications and researches in the field of biomedical science. At present, for combating pathogenic bacteria, we agree that both metabolic inhibition and ROS inhibition are important strategies to inhibit bacterial replication. Researchers have found that the association with specific ZnO NPs geometries interferes with the conformational reorganization of the enzyme β -galactosidase (GAL) (*ACS Nano* **2015**, 9, 9097-9105). It is highly interesting that these biomimetic ZnO NPs with specific morphology could inhibit the growth of MRSA in a shape-specific antibacterial manner. The corresponding comparison of these different antibacterial mechanisms have been added on page 2 in the revised introduction and also shown as follows:

Page2: “Currently, there is an ever-growing need to develop non-antibiotic strategies to fight bacteria without the trigger of drug resistance, and numerous efforts have been paid to this field with the advancements of nanoparticles (NPs), for instance, loading multiple antibacterial molecules into NPs to reduce the formation of drug-resistance⁶, and tuning geometry of NPs to enhance the enzyme inhibition and antibacterial activities⁷. Recently, generating reactive oxygen species (ROS) via chemodynamic, photodynamic, sonodynamic, and enzyme-mimetic materials to combat bacteria has also been explored as extremely promising alternative antibacterial strategies^{8,9}.”

(2) “*The strategy capture and kill shows its efficiency. It seems to me a part of the reason that the Art M are not agglomerating in dispersion. This is the direct consequence of their enhanced dispersability.*”

Response to comment:

Thanks for your good comments. In our manuscript, the Fe-Art M realizes great antibacterial efficiency towards MRSA, and the requested concentration of Fe-Art M can be as low as 8 $\mu\text{g}/\text{mL}$ with H_2O_2 (100 μM), which is very close to the dosage of vancomycin (~ 4 $\mu\text{g}/\text{mL}$). The synthesized Fe-Art M is supposed to act as an artificial macrophage to capture and kill MRSA by the hedgehog topography and massive generation of ROS with its POD-/HPO-mimetic catalytic activities. Yes, we agree with the reviewer that the increased water dispersibility may also facilitate the bacterial capture capability due to this hedgehog topography. To validate that the water dispersibility increased, we have added some additional experiments. Interestingly, it is noticed that Fe-Art M exhibits good dispersibility in diverse

aqueous media, which is much better than the control samples Fe-C-MPs with a smooth surface (Supplementary Figs. 5-6). An earlier report has already demonstrated that the hedgehog structure can efficiently enhance the aqueous dispersibility of particles due to the decreased contact area, trapping of air, autoionization of water, etc. (*Nature*, **2015**, 517, 596-599). Therefore, combined with theoretical and experimental outcomes, we agree with the reviewer that the good water dispersibility of Fe-Art M may facilitate the bacterial capture capability. These data and the corresponding statements have been added on page 4 in the revised manuscript and also in the revised supplementary information, which are also shown as below:

“Interestingly, it is also noticed that the carbonaceous Fe-Art M exhibit good dispersibility in diverse aqueous media than Fe-doped carbonous micro-particles (Fe-C-MPs, smooth surface) (Supplementary Figs. 5-6). An earlier report has suggested that the hedgehog structure can efficiently enhance the aqueous dispersibility of particles due to the decreased contact area, trapping of air, autoionization of water, and etc.²⁷, which may facilitate the bacterial capture capability.”

Supplementary Figure 5. Digital photographs of **a** original solutions, **b** Fe-Art M suspensions, and **c** Fe-C-MPs suspensions for varied durations (including 0, 5, 10, 20, and 30 min). After 5 min, the micrometer-scale Fe-C-MPs all settled, while Fe-Art M can maintain a suspension for about 20-30 min, thus showing that the dispersibility of Fe-Art M (hedgehog topography) is better than that of Fe-C-MPs (smooth surface). PBS: phosphate buffer saline; BSA: bovine serum albumin; FBS: fetal bovine serum; DMEM: Dulbecco's

modified eagle medium. It is noticed that after adding Fe-Art M, the DMEM solution becomes transparent due to its porous structure induced high adsorption capability.

Supplementary Figure 6. The bottom of Fe-Art M and Fe-C-MPs solutions. The duration time is 5 min. Compared to a large amount of sediment in the Fe-C-MPs (smooth surface) group, the Fe-Art M shows almost no precipitant, thus indicating the enhanced dispersibility due to hedgehog topography.

(3) “DFT calculations typically have hard time describing the effect of water media. How did the authors approached this problem here?”

Response to comment:

Thank you for your good comment. First, we are sincerely sorry for the insufficient statement here. As you have known, due to the limitation of theoretical calculation technology, it is indeed a big challenge for DFT calculations to consider the effect of water media. Because of this limitation, almost all the DFT calculations are carried out under vacuum conditions in the current literature. However, it has been found that the results of DFT calculations are in general agreement well with the experimental research results even though the water media is not considered. Yes, of course, we agree with the reviewer that the water media will make the system much more complex; yet, it is believed that the DFT calculations under vacuum conditions can also give a general trend of the experimental results.

At present, researchers in the computing field are still exploring applying DFT calculations in aqueous solutions, while the computation cost and calculation difficulty are extremely high due to the complex environments. In this study, even though the water media is not considered, the results of DFT calculations are consistent with our experimental data; meanwhile. Since it is extremely hard for us to calculate the enzyme-mimetic catalytic process in aqueous media, we hope that the reviewer can understand this difficulty and allow us to waive the further demonstration on this issue. And in our future researches, we will carefully consider your suggestions if there are possible methods that can be used.

Reviewer #2:

“The authors have prepared a hedgehog-like micrometer-sized particle and evaluated the potential of this particle as disinfectant for promoting wound healing. I don’t think this work is suitable for publication in Nature Communications due to the following major concerns and inconsistency and would recommend submission somewhere else like Chemical Communications.”

Response to the general comment:

Thanks a lot for your important comments and helpful suggestions to further improve the quality of this manuscript. Based on your comments and the other reviewers’ suggestions, we have performed more systematic experiments and further polished all the contents. All the necessary data have been added to support our claims, and all the questions and concerns have been well-addressed in the revised manuscript and revised supplementary information. Therefore, we believe that the quality of this paper has been significantly enhanced. We hope that you can agree with this assessment, and we thank you again for your great efforts.

(1) *“The scientific question or technical limitation which this work is to address is not clear. On page 3 in the introduction section, the authors stated that “...when treated at a very low concentration, such enzyme-mimetic catalysts present insufficient bacterial killing activities, while utilizing antibacterial materials at high concentrations will always lead to a severe concern on biocompatibility...”. But, unfortunately, the hedgehog-like particle reported in this work is cytotoxic to mammalian cells as well, according to Supplementary Figure 32, which though a qualitative assay revealed dead cells (in red) to appreciable ratio no matter whether or not external H₂O₂ was added. In fact, the authors need to quantitatively and qualitatively characterize the antibacterial activity and cytotoxicity of their particle.*

As summarized by the authors in Supplementary Table 4, there are already quite many reports on nanozymes and I believe that there more nanozymes in the literature than what had been summarized here in Supplementary Table 4. Compared with the previously reported nanozymes, what advantages or uniqueness does the hedgehog particle reported in this manuscript have? If it is just another particle that has catalytic activity, generates ROS, and kills bacteria as do so many of its prior counterparts have already done, then its novelty is limited and it should not be acceptable for publication in Nature Communications.”

Response to comment:

Thanks for your important comments. We will carefully reply to your question in two parts.

For the biosafety issues: First, although great efforts have been paid to the exploration and expansion of nanotechnology and nanomaterials, the biggest biosafety problem of nanomaterials in biomedical fields is still the teratogenic and carcinogenic risks caused by long-term exposure or accumulation of nanomaterials. And the biosafety problem would become more serious when the treatment needs longer contacting time and larger dosage. To solve the concerns on biosafety issues of antibacterial materials for wound disinfection, we believe that greatly reducing the needed dosage and contacting time of materials is the best way.

In our manuscript, we proposed a new and extremely efficient antibacterial strategy by mimicking the “capture and killing” process of macrophages to combat bacteria, which can significantly improve the catalytic bactericidal performances and maintain limited cellular toxicity. The experimental studies and theoretical calculations reveal that the synthesized artificial macrophage can efficiently capture and kill MRSA by the hedgehog topography and massive generation of $\bullet\text{O}_2^-$ and HClO with its $\text{Fe}_2\text{N}_6\text{O}$ catalytic centers. The synthesized artificial macrophage exhibits an extremely low minimal inhibition concentration (8 $\mu\text{g}/\text{mL}$ Fe-Art M with H_2O_2 (100 μM)) to combat MRSA and can rapidly promote bacteria-infected wound healing of rabbit skin, which is comparable to the treatment efficiency of vancomycin (~ 4 $\mu\text{g}/\text{mL}$).

By comparing with other reported nanoagents with different antibacterial actions, including ROS generation, ions release, photothermal, biofilm penetration, and membrane disruption, as shown in Supplementary Table 5, our Fe-Art M achieves the lowest antibacterial concentration benefiting from its unique and efficient bioinspired “capture and killing” process. In summary, our research not only increases the antibacterial performance but also significantly reduces the needed dosage of biocatalysts, thereby dramatically lowering the teratogenic and carcinogenic risks and improving the biosafety.

Another unique advantage of applying our Fe-Art M for wound disinfection is that these materials are not only highly efficient for ROS generation but also present highly stable $\text{Fe}_2\text{N}_6\text{O}$ catalytic structures, as disclosed in Fig. 4d and Supplementary Table 4. For example, the earlier reported vanadium oxides also present efficient antibacterial activity by mimicking HPO to generate HClO (*Nat. Nanotechnol.* **2012**, **7**, 530-535); however, the release of unstable V ions may cause potential teratogenic and carcinogenic risks. In our study, there will be no risks of releasing unstable metal ions.

For the wound disinfection studies, we believe that the short-term effects on ordinary cells of antibacterial materials are not a critical issue, which may not lead to serious concerns on biocompatibility.

Actually, the treatment of skin wound infection is highly different from many other diseases, for instance, antitumor. The complete disinfection treatment must be achieved within a short time since the bacteria reproduce very fast. If the sterilization is not complete, the infection may happen again, which will cause huge trouble for the patient. Therefore, we need to design the materials with extremely high ROS production capability and bacteria-killing efficiency. Only through this way, we can make balance on the efficient bacterial sterilization and low tissue toxicity.

For cellular toxicity, we agree with the reviewer, of course, some tissue cells will definitely be killed during the disinfection process. But this won't be a big problem, as long as all the bacteria are cleaned up, the tissue cells will soon recover and regenerate; however, if the bacteria are not totally eradicated, a secondary infection will happen due to the high reproduction speed of bacteria. What's more, it is undoubtedly that many antibacterial materials are toxic when a high dosage is used, including many frequently used drugs, polymers, and nanomaterials. In our disinfection process, all materials are only exposed to the surface skin, no blood vessels and deep tissues are involved; after the bacteria are killed, the materials can be washed away immediately to reduce the potential toxicities to cells, blood, and tissues, and the risks of potential cellular uptake and long-term cumulative toxicity caused by endocytosis are extremely low. Besides, we believe that the most important issue in antibacterial studies of wound disinfection should be the realization of rapid and complete sterilization; as long as the materials are stable during the treatment.

Furthermore, we are very grateful for your helpful suggestions on quantitatively and qualitatively characterizing the antibacterial activity and cytotoxicity of Art Ms, and we have added these data in the revised manuscript and supporting information to improve the quality of this paper further. As it can be seen, no MRSA colonies are observed for the Fe-Art M+H₂O₂ group, which means the MRSA has been completely eradicated by our materials (Supplementary Fig. 50).

Additionally, we have also added the living/dead fluorescence images and CCK-8 assay of different samples to calculate cell viability, and we found that in the absence of H₂O₂, the death ratio of the cell is about 10%, which should be attributed to the spiky structure and oxidase-like activity, i.e., catalytic transformation of oxygen into ROS by activation and cleavage of the O-O bond. Besides, when adding H₂O₂, although the generation of ROS would kill a certain amount of tissue cells, the cell populations could recover quickly within a few days (Supplementary Figs. 42-45).

Supplementary Table 5. Comparison of the antibacterial efficiency and concentrations between Fe-Art M+H₂O₂ and other antibacterial agents.

Catalysts	External condition	Antibacterial action	Antibacterial concentration	Antibacterial effects	Ref.
Fe-Art M	H ₂ O ₂	Capture and ROS generation	8 µg/mL	100% (MRSA)	This work
Vancomycin	/	/	4 µg/mL	100% (MRSA)	This work
PPPC-1	Ultrasound	ROS generation	0.04 mmol/L	100% (MRSA)	¹³
Cu ₂ MoS ₄	Near-infrared II (NIR-II) light	NIR-II light enhanced POD/OXD	40 µg/mL	~100% (MDR S. aureus)	¹⁴
Fe-BBP	Ultrasound, H ₂ O ₂	ROS generation	200 µg/mL	~100% (MRSA)	¹⁵
P5-Polymeric NPs	/	Biofilm penetration and eradication	84 µg/mL	~100% (MRSA)	¹⁶
HPEM	/	Membrane disruption	2.3 g/mL	99.03% (MRSA)	¹⁷
ANVs nanocapturer	Ultrasound	Capture and ROS generation	200 µg/mL	80% (MRSA)	¹⁸
Fe ₃ O ₄ /CNT/Gent	Microwave	Capture and thermal ablation	1 mg/mL	99.72% (MRSA)	¹⁹
PDGu(7)-b-PBLK(13)	/	Membrane disruption and	256 µg/mL	99% (MRSA)	²⁰

		interface weakening effects			
ZnPB-3	NIR	Photothermal ablation and ion release	200 µg/mL	99.66% (MRSA)	²¹
Gly-POX ₂₀	/	Membrane disruption and ROS generation	25 µg/mL	100% (MRSA)	²²
MoS ₂ /rGO VHS	Light, H ₂ O ₂	Bacterial capture and ROS generation	50 µg/mL	100% (S. aureus)	²³
V-POD-M	H ₂ O ₂	Bacterial capture and ROS generation	16 µg/mL	100% (S. aureus)	²⁴
PEG-MoS ₂ NFs	NIR, H ₂ O ₂	Photothermal activities and ROS generation	100 µg/mL	99% (S. aureus)	²⁵
TRB-ZnO@G	NIR	Triple antibacterial activities (chemical, mechanical and photothermal)	50 µg/mL	100% (S. aureus)	²⁶
C-Zn/Ag	NIR	Ions release and photothermal activities	160 µg/mL	100% (S. aureus)	²⁷

PPPC: polymer-peptide-porphyrin conjugate; Fe-BBP: Fe³⁺-BiOBr-polyethylenimine; HPEM: HCHO-PGE-MXene@PDA; ANVs: MAb-piloting nanovesicles; PDGu(7)-b-PBLK(13): poly(amido-D-glucose)-block-poly(beta-L-lysine); ZnPB-3: zinc-doped Prussian blue; Gly-POX₂₀: poly(2-oxazoline); MoS₂/rGO VHS: molybdenum disulfide/rGO vertical heterostructure; V-POD-M: virus-like peroxidase-mimic; PEG-MoS₂ NFs: polyethylene glycol functionalized molybdenum disulfide nanoflowers; TRB-ZnO@G: thermally responsive brushes-ZnO-doped carbon on graphene; C-Zn/Ag: Ag-doped carbonized ZIF nanocomposite.

Fig. 4d Comparison of the TON and V_{\max} values with reported enzyme-mimetic catalysts. NPs: nanoparticles; MOFs: metal-organic frameworks.

Supplementary Figure 50. a Agar plate digital photographs after treating MRSA with Art M materials. **b** MRSA killing ratio of different treatments. n=3, data are presented as mean \pm SD. Statistical significance is assessed by unpaired Student's two-side t-test compared to the control groups. (***)P<0.001).

Supplementary Figure 42. Representative live/dead cell staining (green: live, red: dead) for bare L929 and samples co-cultured L929. Reaction condition: [catalyst]=10 μ g/mL, [H₂O₂]= 100 μ M) for 24, 48, and 72 h.

Supplementary Figure 43. a CCK-8 assay for co-culture of bare L929 cells and Art M samples. **b** Cell viability after co-culture of bare L929 cells and Art M samples. (n=3 per group, data are presented as mean \pm SD). Statistical significance is assessed by Student's two-side t-test. (*P<0.05, **P<0.01).

Supplementary Figure 44. Representative live/dead cell staining (green: live, red: dead) for various samples after co-cultured with L929 cells.

Supplementary Figure 45. a CCK-8 assay for co-culture of bare L929 cells and Art M samples with or without H_2O_2 . **b** Cell viability after co-culture of bare L929 cells and Art M samples with or without H_2O_2 . (100 μM H_2O_2 , $n=3$ per group, data are presented as mean \pm SD). Statistical significance is assessed by Student's two-side t-test. (* $P<0.05$, ** $P<0.01$).

Regarding the comment on novelty and new findings of this study:

We thank you for this important question. The main motivation of this work originates from two aspects: (1) designing highly stable hedgehog micro-sized carbonaceous particles to mimic the topography of macrophage for localized bacterial capture, (2) precisely engineering single-atom catalytic sites to mimic the heme-structure of HPO for ROS production. As we have already discussed in the introduction part and listed in Supplementary Table 4; indeed, there are already some reports that have engineered enzyme-mimics or nanozymes for antibacterial applications. Nevertheless, when treated at a very low concentration, such enzyme-mimetic catalysts present insufficient bacterial killing activities, while utilizing antibacterial materials at high concentrations will always lead to a severe concern on biocompatibility. Thus, developing more effective ROS-based enzyme-mimetic strategies to fight drug-resistant bacteria is of great importance in current material and biomedical sciences. The human immune system has provided us an ideal example to overcome this bionic challenge; it has been clarified that macrophage plays a critical role in disinfection and wound healing. By simultaneously mimic the “capture and killing” process of macrophages to combat bacteria, for the first time, we proposed to design a hedgehog artificial macrophage with atomic-catalytic centers to combat MRSA, especially integrated with

heme-mimetic catalytic sites, which definitely presents a very new and unique design to fabricate antibacterial materials via bionic disinfection strategies. To reply to your concerns on the novelty of this proposed new material and antibacterial system, we would like to make further explanations with the following reasons:

1) Of course, enzyme-mimics or nanozyme is a very hot research area; there will always be some good publications that have been reported. But many of the current reported enzyme-mimics or nanozymes are designed for antitumor systems. Among these researches, the focuses are mainly on improving tumor targeting by introducing ferritin or wrapping erythrocyte membrane, solving the problem of tumor hypoxia, and realizing functional cascades reactions (*Nat. Commun* **2018**, 9, 11; *Nat. Commun* **2019**, 10, 240; *Nat. Nanotechnol.* **2021**. <https://doi.org/10.1038/s41565-021-00910-7>, *Sci. Adv.* **2020**; 6: eabb1421).

2) Among recently reported ROS generation enzyme-mimetic materials for bacterial disinfection, including nanoliposome, metal-organic frameworks, and inorganic materials, they usually present insufficient bacterial killing activities when treated at a very low concentration. Developing more effective ROS-based enzyme-mimetic strategies to fight drug-resistant bacteria is of great importance in current material and biomedical sciences. However, materials that have superhigh ROS-catalytic activity and bacterial capture capabilities are rarely reported. One recent report has designed a graphene-isolated PtCo nanomaterial for bacterial inhibition, but this material only has oxidase-mimetic property (*Nat. Commun* **2021**, 12, 2002). Meanwhile, when these materials are applied for antibacterial applications, they usually lack strong direct interactions with bacteria. Since ROS quench very fast, the poor interaction between materials and bacteria will lead to low antibacterial efficiency, especially at a low dosage of materials. In our design, the synthesized hedgehog artificial macrophage can efficiently capture and kill MRSA by the hedgehog topography and localized massive generation of $\bullet\text{O}_2^-$ and HClO with its unique $\text{Fe}_2\text{N}_6\text{O}$ catalytic centers. Meanwhile, it is also noticed that all the carbonaceous Art M exhibit good dispersibility in diverse aqueous media; the earlier report has suggested that the hedgehog structure can efficiently enhance aqueous dispersibility of particles due to the decreased contact area, trapping of air, autoionization of water, and etc. (*Nature*, **2015**, 517, 596-599), which may facilitate the bacterial capture capability (Supplementary Figs. 5-6).

3) In this study, we achieve the construction of hedgehog artificial macrophages to combat bacteria for the first time by simultaneously mimicking the “capture and killing” process of macrophages and integrating the peroxidase-mimetic ROS-catalytic sites. Our material could first efficiently trap MRSA by

its spiky topography and bioadhesive carbonaceous structure and then generate abundant localized ROS to kill them, which solves the short lifetime problem of ROS, thus significantly decreasing the needed dosage. The synthesized artificial macrophage exhibits an extremely low minimal inhibition concentration (8 $\mu\text{g}/\text{mL}$ Fe-Art M with H_2O_2 (100 μM)) to combat MRSA and rapidly promote bacteria-infected wound healing of rabbit skin, which is comparable to the treatment efficiency of vancomycin.

4) Besides the high intrinsic oxidase-mimetic activity, the synthesized Fe-Art M was filled with atomically dispersed $\text{Fe}_2\text{N}_6\text{O}$ catalytic active centers; it will present POD- and HPO-like activities simultaneously in the presence of H_2O_2 and Cl^- , thus generating O_2^- and HClO efficiently. Due to the strong coordination and fully exposed atomic catalytic sites, the Fe-Art M exhibits the highest turnover number (TON) value among most of the current established POD-mimics, which suggests that the $\text{Fe}_2\text{N}_6\text{O}$ catalytic sites are very efficient in conversing H_2O_2 substrate (Fig. 4d).

5) Furthermore, in this study, besides the new experimental findings, we have also carefully elucidated the detailed POD-like and HPO-like ROS-catalytic pathways of Fe-Art M by using the DFT calculations to simulate the $\bullet\text{O}_2^-$ and HClO-generating processes. This is the first time that the $\bullet\text{O}_2^-$ and HClO-generating processes have been calculated by the DFT method. In the earlier study, only the $\bullet\text{OH}$ radical generation process has been simulated. We have carefully proposed different potential reaction pathways and compare them in detail by electron density difference and free energy diagrams to reveal the most reasonable POD-like and HPO-like ROS-catalytic reaction pathways.

Based on the above-detailed clarification on the novelty of this study, it can be easily seen that this submission shows no similarity on the main motivation and concept to any earlier publication. The generated localized ROS via strong bacterial capture capabilities of hedgehog structures has not been reported before. The POD-like and HPO-like ROS-catalytic reaction pathways are summarized for the first time by this study. Therefore our antibacterial material definitely is not just another particle that has the catalytic activity of ROS generation for bacterial eradication. Therefore, we believe that this is a completely new antibacterial system with solid experimental and theoretical validations; there is no such a unique design in all previous reports. To sum up, we believe that this work shows a good novelty with essential new findings on designing biomimetic antibacterial materials for localized ROS-catalytic sterilization. We predict that this inexpensive, durable, and highly ROS-catalytic active Fe-Art M with “capture and killing” action will not only provides a promising broad-spectrum therapy for non-antibiotic

wound disinfections but also shows enormous possibilities to serve as antibacterial agents for many other biomedical applications.

With all these clarifications and statements, I hope that you will agree with us on the novelty assessment of this work. We thank you again for your great efforts in reviewing our manuscript.

Some of the above discussions have already been clearly presented or added on page 2-4 in our revised manuscript:

“Recently, generating reactive oxygen species (ROS) via chemodynamic, photodynamic, sonodynamic, and enzyme-mimetic materials to combat bacteria has also been explored as extremely promising alternative antibacterial strategies^{8,9}. Among diverse ROS generation materials, the enzyme-mimetic catalysts, including nanoliposome¹⁰, metal-organic frameworks¹¹⁻¹³, and inorganic materials¹⁴, have gained tremendous popularity by catalyzing hydrogen peroxide (H₂O₂) to generate ROS via the existing natural pathway. Nevertheless, when treated at a very low concentration, such enzyme-mimetic catalysts present insufficient bacterial killing activities, while utilizing antibacterial materials at high concentrations will always lead to a severe concern on biocompatibility. Thus, developing more effective ROS-based enzyme-mimetic strategies to fight drug-resistant bacteria is of great importance in current material and biomedical sciences.

The human immune system has provided us an ideal example to overcome this bionic challenge; it has been clarified that macrophage plays a critical role in disinfection and wound healing¹⁵⁻¹⁷. The macrophage can efficiently eradicate pathogenic bacteria by successively capturing, phagocytosing, and killing actions. The macrophage exhibits a rough surface with microfold, prominences, and recognition receptors, which facilitate the detection and capture of bacteria. After phagocytosis, various peroxidases (POD) in the lysosome can produce abundant localized ROS to kill phagocytosed bacteria rapidly^{18,19}. Recently molecular biology reveals that these ROS-generation enzymes mainly consist of heme-containing catalytic sites, which can generate highly toxic hypochlorous acid (HClO) and other types of ROS in the presence of Cl⁻ and H₂O₂²⁰⁻²². Though V₂O₅ nanorods have been reported to show excellent haloperoxidase (HPO)-mimetic production activity of HClO, the potential cellular toxicity and poor bacterial capture ability of V₂O₅ may limit their application in clinical treatment²³. Thus, it is a crucial challenge to develop ROS catalytic materials that can simultaneously mimic the “capture and killing” process of macrophages to combat bacteria, especially integrated with heme-mimetic catalytic sites.

To address this challenge, we present here the design of a hedgehog artificial macrophage with heme-mimetic atomic-catalytic centers to combat MRSA. The main motivation of this work originates from two aspects: (1) designing highly stable hedgehog micro-sized carbonaceous particles to mimic the topography of macrophage for localized bacterial capture, (2) precisely engineering metal-N_x-C_y based single-atom catalytic sites to mimic the heme-structure (Fig. 1a); here the Fe-N_x-C_y is the primary candidate, while the V-N_x-C_y is synthesized for comparison due to the natural widely existence of Fe-HPO (Fig. 1b) and V-HPO (Fig. 1c) for HClO production. The experimental studies and theoretical calculations reveal that the synthesized artificial macrophage can efficiently capture and kill MRSA by the hedgehog topography and localized massive generation of •O₂⁻ and HClO with its Fe₂N₆O catalytic centers. The synthesized artificial macrophage exhibits an extremely low minimal inhibition concentration (8 μg/mL Fe-Art M with H₂O₂ (100 μM)) to combat MRSA and can rapidly promote bacteria-infected wound healing of rabbit skin, which is comparable to the treatment efficiency of vancomycin.”

(2) *“The authors stated that their hedgehog-like particle can capture the bacteria and then kill the as-captured bacteria on contact. A question which the authors failed to address with experimental evidences is why their particle can capture the bacteria. They claim that their particle has rough surface and, as a result, should be able to capture bacteria nearby. Such a claim is not persuasive and needs support from experimental evidences. I would suggest the authors to prepare particles of similar size and surface chemistry as the hedgehog-like one here but instead with smooth surface and check whether bacterial cells adsorb onto this smooth-surfaced counterpart.”*

Response to comment:

Thank you for the good comments, and we are sorry for the insufficient statements and control samples to clearly demonstrate our claims. In this study, the strong capability to capture the bacteria of hedgehog artificial macrophages should be attributed to its spiky topography and bioadhesive carbonaceous structure. We have carried out systematic studies based on the reviewers' suggestions. We first synthesized or obtained several different comparison samples with similar particle sizes as that of Art M, such as Fe-doped carbonous micro-particles (Fe-C-MPs, smooth surface, Supplementary Figs. 11), polystyrene microspheres (smooth surface), and SiO₂ microsphere (slightly rough surface).

It can be observed that all types of Art Ms can efficiently capture abundant bacteria with no significant difference, as demonstrated by the SEM and fluorescence microscopy images (Figs. 2f-j and

Supplementary Figs. 12-14). The results of Fe-C-MPs and SiO₂ microspheres indicate that both the carbonous surface and slightly rough surface exhibit limited bacterial capture capability (Supplementary Figs. 15, 16); while, the polystyrene microsphere with a smooth surface shows nearly no bacterial capture capability (Supplementary Figs. 17). Thus, these added new data can directly demonstrate the concept that these hedgehog artificial macrophages can be applied to capture bacteria via their spiky topography and bioadhesive carbonaceous structure.

For the strong interactions between bacteria and carbonaceous structure, we have summarized these interactions in our earlier reviews, such as *Adv. Mater* **2017**, 29, 1602547, *Chem. Rev* **2017**, 117, 1826-1914. In brief, these interactions are derived from the abundant proteins on the bacterial surface, which can interact with the carbonaceous structure. By applying the molecular dynamic simulations, it has been depicted that the protein adsorption on carbon surface is mainly driven by the π - π stacking due to aromatic protein residues; meanwhile, it is also suggested that the hydrophobic interactions of protein structure can also promote the adsorption process. Our data also validate that a rough carbon surface will provide more intensive bacterial adhesion at the carbon interfaces.

In order to provide proof that Fe-Art M can capture all types of bacteria via its spiky topography and bioadhesive carbonaceous structure, we also added the bacterial capture capability on the extended-spectrum β -lactamase-producing *E. coli*. The data show that the Art Ms can also efficiently capture the bacillus, thus demonstrating its universal capture ability to bacteria.

The corresponding discussion and new results have been added on page 6 in the revised manuscript and pages 14-18 in the revised supplementary information:

Page 6:“To directly demonstrate the concept that these hedgehog artificial macrophages can be applied to capture the bacteria via its spiky topography and bioadhesive carbonaceous structure²⁹, we have incubated MRSA with different Art Ms and comparison samples (Fe-C-MPs, smooth surface, Supplementary Fig. 11), polystyrene microspheres (smooth surface), SiO₂ microsphere (slightly rough surface). It can be observed that all types of Art Ms can efficiently capture abundant bacteria with no significant difference, as demonstrated by the SEM and fluorescence microscopy images (Figs. 2f-j and Supplementary Fig. 12-14). The results of Fe-C-MPs and SiO₂ microspheres indicate that both the carbonous surface and slightly rough surface exhibit limited bacterial capture capability (Supplementary Figs. 15, 16); while, the

polystyrene microsphere with smooth surface shows nearly no bacterial capture capability (Supplementary Fig. 17).”

Supplementary Figure 11. SEM images of **a** ZIF-8 large crystal, **b** carbonized ZIF-8 (C-ZIF-8), and **c** Fe-C-MPs. **d** XRD patterns of ZIF-8 and Fe-C-MPs. **e** Comparison of curve-fitted high-resolution XPS N 1s spectra of the C-ZIF-8 and Fe-C-MPs. **f** High-resolution Fe 2p XPS spectra for Fe-C-MPs. The SEM images of ZIF-8, C-ZIF-8, and Fe-C-MPs demonstrates that they share a similar size as Art Ms. As can

be seen in the XRD and XPS results, the metal Fe in Fe-C-MPs exists in the form of coordination structure due to no peaks of iron oxides and the peak shift after the coordination with Fe, indicating the chemical structure of Fe-C-MPs is similar to that of Fe-Art M.

Supplementary Figure 12. SEM images of Art M, V-Art M, and Fe-Art M treated with MRSA.

Supplementary Figure 13. Fluorescence microscopy images of Control and Art M when incubated with MRSA.

Supplementary Figure 14. SEM images of bacterial capture capability of Art Ms when incubated with extended-spectrum β -lactamase-producing *E. coli*.

Supplementary Figure 15. SEM images of **a** MRSA and **b** extended-spectrum β -lactamase-producing *E. coli* after incubation with C-ZIF-8 and Fe-C-MPs, which indicate that the carbon material with a smooth surface shows weak bacterial capture capability.

Supplementary Figure 16. SEM images of **a** MRSA and **b** extended-spectrum β -lactamase-producing *E. coli* after incubation with SiO₂ microspheres, which indicates that the slightly rough surface exhibits relatively limited bacterial capture capability.

Supplementary Figure 17. SEM images of **a** MRSA and **b** extended-spectrum β -lactamase-producing *E. coli* after incubation with polystyrene microspheres, which indicate that the polymeric material with a smooth surface shows nearly no bacterial capture capability.

(3) “Throughout the manuscript, the authors mentioned three different types of enzyme-mimicking activities their particle exhibited, which are oxidase-like on page 6, peroxidase (POD)-like on page 9, and haloperoxidase (HPO)-like also on page 9 (and of course later on other pages). Oxidase, peroxidase, and haloperoxidase use different substrates and correspond to different ROS-generating reaction routes. On page 6, when the authors claimed their particle to be able to convert O_2 into ROS, they used Supplementary Figure 11 to support this claim. Nevertheless, Supplementary Figure 11 did not convey any information on whether O_2 is a substrate for the ROS generation mediated by this hedgehog-like particle; instead, it only shows that there is ROS production at pH 4.5, a very acidic pH.

In fact, in both the in vitro antibacterial assays and the rabbit model study, the authors added H₂O₂ (10 mM in the former while 100 mM in the latter) externally in addition to their particle. If their particle is oxidase-like, why should the authors spend extra efforts to add external H₂O₂? If externally added H₂O₂ is necessary, then the potential administration of their hedgehog particle is superficial sites like wounded skin; otherwise, where can such high H₂O₂ dose come from naturally or endogenously?

According to the animal study in this work, the hedgehog-like particle did not offer better therapeutic efficacy than vancomycin. Then why should the audience care about using such an inorganic particle to do a job which an available antibiotic can well accomplish?"

Response to comment:

Thanks for your important and helpful comments to improve the quality of this manuscript. We will carefully reply to your comments in the following three parts: 1) oxidase-like catalytic activity, 2) why H₂O₂ is needed, and 3) the importance of hedgehog artificial macrophage in combating bacteria.

For the **1) oxidase-like catalytic activity**: First, as demonstrated by our experiments, in the absence of H₂O₂, the dead MRSA captured by the hedgehog Fe-Art M was caused by the spiky structure and oxidase-like catalytic ROS generation. As presented in the revised Supplementary Figure 18, we have analyzed the intrinsic oxidase-mimetic activity of these Fe-Art M, which suggests that the Fe-Art M exhibits very high catalytic transformation of oxygen into ROS via activation and cleavage of the O-O bond as demonstrated by earlier literature (*Sci. Adv.* **2019**, 5, eaav5490), based on this literature, the Fe-N bond is the catalytic center to activate the oxygen to generate ROS. Other recent reports have also shown that the oxidase-mimetic activity-induced ROS can be applied for antibacterial applications (*Nat. Commun.* **2021**, 12, 2002). This good contact killing activity of Fe-Art M preliminarily supports the proposed concept for the bacterial "capture and killing" function of hedgehog artificial macrophage. However, the antibacterial capability from the oxidase-like ROS generation is very limited, which is not the focus of this study; we focus on the POD-like and HPO-like activities for combating bacteria. Therefore, we didn't put the data of oxidase into the main text, hoping that the reviewer can agree with this assessment. Furthermore, we agree with the reviewer that solid evidence is needed to prove whether O₂ is a substrate for the ROS generation for the Fe-Art M; therefore, we have added the UV-vis absorption spectra of TMB after catalytic oxidation by Fe-Art M in O₂-saturated, air-saturated, and Ar-saturated buffers, as shown in Supplementary Figure 18, which give clear evidence that the Fe-Art M can efficiently catalyze O₂ substrate into ROS.

Supplementary Figure 18. The oxidase-like activity: **a** Typical absorption spectra of TMB after catalytic oxidation in pH 4.5 acetate buffer. **b** Ultraviolet-visible (UV-vis) absorption spectra of TMB after catalytic oxidation in pH 4.5 by the Fe-Art M in O₂-saturated, air-saturated, and Ar-saturated sodium acetate-acetic acid buffer.

Fig. 2 g The calculated number of MRSA captured by different Art M. The inset of **g** is a magnified captured bacteria by the spikes. **h** Calculated live/dead bacteria ratio based on the fluorescence results of **i** V-Art M and **j** Fe-Art M, pH= ~5.8; data were obtained from three fluorescent microscopy images with scope 1 mm×1 mm, and presented as mean ± SD. Statistical significance is assessed by Student's two-side t-test. (*P<0.05, **P<0.01, ***P<0.001).

2) why H₂O₂ is needed? In this study, we propose to design a hedgehog artificial macrophage with atomic-catalytic centers to combat MRSA by mimicking the “capture and killing” process of macrophages. This contact-killing activity derived from oxidase-like ROS generation of Fe-Art M can only achieve very limited antibacterial capability due to the limited amount of O₂ in the bacterial growth media. Therefore, our results show that the oxidase-like activity-induced ROS is not sufficient to kill all bacteria in the media (Fig. 2h). Therefore, to achieve efficient “capture and killing” process, we didn’t focus on the oxidase-like ROS generation; we found that the POD-like and HPO-like activities are much more efficient for combating bacteria by utilizing the H₂O₂ and Cl⁻ as substrates, which can initiate the POD-like and HPO-like activities by catalyzing the H₂O₂ and H₂O₂+Cl⁻ to generate strong and high concentration ROS (\bullet O₂⁻ and OCl⁻) for efficiently eradication of MRSA.

About the H₂O₂ concentration, we are sorry that we have made a misleading description in the supporting information; we forget to transfer the used H₂O₂ concentration into the final H₂O₂ concentration of the aqueous system. In our study, the final H₂O₂ concentration should be 100 μM for both *in vitro* and *in vivo* experiments. The 100 μM H₂O₂ concentration is a general standard value taken by some earlier antibacterial studies, for instance, the *ACS Nano* **2016**, 10, 11000-11011. We found that the 100 μM H₂O₂ concentration is the best concentration for this study, which shows limited effects on the bacteria and cells without catalyst (Fig. 6b and Supplementary Fig. 42-45), while, after adding the Fe-Art M, it shows very quick disinfection capability (Fig. 6b-e).

For this study, we agree with the reviewer that it is not possible to find high H₂O₂ concentrations from natural or endogenous environments. Therefore, this new antibacterial material is mainly used for skin and wound disinfection and other external sterilization applications. We have taken the skin wound disinfection as an example to illustrate the excellent bacterial eradication ability and application potentials of such a new antibacterial material for bionic and non-antibiotic disinfection. In wound disinfection, the H₂O₂ concentration at 100 μM is a frequently used concentration with good balance on the catalytic ROS generation and biosafety concern, which we think is suitable for future potential disinfection applications. Besides the wound disinfection, we think that these Fe-Art M can also be applied in some other areas, for instance, marine biofouling and face masks. In our report, the Fe-Art M can efficiently mimic HPO to generate ClO⁻, which can even work at pH 8, thus making it potential for marine biofouling (*Nat. Nanotechnol.* **2012**, *7*, 530-535). Of course, much more experiments are needed in the future before we can clearly disclose their application potentials.

3) for the importance of hedgehog artificial macrophage in combating bacteria:

In general, why the non-antibiotic disinfection strategies and materials are important: Worldwide, the patients' number related to refractory wounds is very high, among these refractory wounds, the deep wounds, burns, scalds, diabetic feet, and skin ulcers have large wound areas and long healing process. The refractory wounds can be easily infected by bacteria which results in strong inflammatory responses and severe skin necrosis (*Nature*, **2016**, 529, 336; *Nat. Biomed. Eng.* **2018**, 2(2), 95-103). Once the bacterial infection occurs on the wound, a large dose of oral or external use of antibiotics is required. However, long-term usage of antibiotics might cause drug resistance, which will then increase the difficulty of wound healing. It often requires repeated debridement, anti-infection, and anti-inflammatory treatments for a very long time. In severe cases, bloodstream infections may occur, thus leading to systemic sepsis, amputation, and even death; these all have brought substantial economic burdens and pain to patients. According to relevant data from the World Health Organization (WHO), there are as many as 700,000 people die from drug-resistant bacterial infections each year worldwide. The British Antibacterial Evaluation Board estimates that by 2050, 10 million people will suffer from infections caused by resistant bacteria worldwide. The use of antibiotics in hospitalized patients with refractory wounds is very high; there are serious antibiotic abuse and drug-resistant bacterial infections (*Primary Care: Clinics in Office Practice*, **2018**, 45, 467-484; *Adv. Mater.*, **2019**, 31, 1806024). Therefore, there is an ever-growing need to develop safe and efficient non-antibiotic strategies and antibacterial materials to fight bacteria without the trigger of drug resistance, which is very important to the current clinical treatment of refractory wounds.

For the importance of our hedgehog particle compared to vancomycin: The synthesized artificial macrophage exhibits an extremely low minimal inhibition concentration (8 $\mu\text{g/mL}$ Fe-Art M with 100 μM H_2O_2) to combat MRSA and can rapidly promote bacteria-infected wound healing of rabbit skin, which shows similar antibacterial efficiency to vancomycin. However, vancomycin is a kind of antibiotic; its overuse may promote the formation of vancomycin-resistant bacteria; for example, vancomycin-resistant MRSA is emerging (*Microbial Pathogenesis*, **2020**, 149, 104537). Therefore, it is extremely important to develop alternative and non-antibiotic strategies to fight MRSA in wound healing treatments. Furthermore, antibiotics have their inherent limitations; for instance, vancomycin can't eradicate gram-negative bacteria (for example, drug-resistant *E. coli*). However, in the presence of H_2O_2 , our Fe-Art M shows broad antibacterial performances for both gram-positive and gram-negative bacteria via its efficient "capture and killing" capability. To support this conclusion, we have added some new data by using the extended-

spectrum β -lactamase-producing *E. coli* as a representative gram-negative bacteria; our materials are still very efficient to these gram-negative bacteria, which can be seen in Supplementary Figure 14, 50. Therefore, the Fe-Art M can be utilized to sterilize all types of bacteria on infectious wounds, such as burned wounds, diabetic feet, chronic ulcers of the skin, and even for surface sterilization and antifouling, which is more efficient than the vancomycin; thus, we believe that the proposed new material and efficient strategy will gain great attention in broad research areas.

Supplementary Figure 14. SEM images of bacterial capture capability of Art Ms when incubated with extended-spectrum β -lactamase-producing *E. coli*.

Supplementary Figure 50. a Agar plate digital photographs after treating extended-spectrum β -lactamase-producing *E. coli* with Art M materials. **b** Extended-spectrum β -lactamase-producing *E. coli* killing ratio of different treatments. n=3, data are presented as mean \pm SD. Statistical significance is assessed by unpaired Student's two-side t-test compared to the control groups. (***)P<0.001). The almost no colony counting of Fe-Art M+ H₂O₂ group means its antibacterial effect toward Extended-spectrum β -lactamase-producing *E. coli*.

(4) “The authors claimed that their hedgehog-like particle produce $\bullet\text{O}_2^-$ and HClO as the ROS species based on their calculations. They need do experiments to examine whether their particle does produce these ROS species and whether there are other ROS species it produces.”

Response to comment:

Thanks a lot for these very helpful comments. To make all the conclusions in this manuscript more solid, we have added the corresponding experiments to verify the ROS species clearly. Since our study mainly focuses on the POD-like and HPO-like catalytic ROS production, therefore, in the following section, we will reply to your comments from these two aspects to reveal the ROS species.

For the POD-like catalytic process, the chromogenic reaction, EPR spectra, and fluorescence reaction were utilized to confirm that the $\bullet\text{O}_2^-$ is the ROS product of Fe-Art M when serving as POD-mimics to catalyze the H₂O₂. First, the TMB method is used to confirm the generation of radicals in the presence of Fe-Art M and H₂O₂, the occurrence of oxTMB (intensity of 652 nm) was detected by UV-vis spectra (Fig. 4a). Then, we take the EPR method to distinguish the detailed ROS species of Fe-Art M; it displays the special signal characteristic of DMPO- $\bullet\text{O}_2^-$ complex adduct (Fig. 4e), and HE probe is also taken to detect the $\bullet\text{O}_2^-$ at \sim 650 nm, the intensity of $\bullet\text{O}_2^-$ can be detected by fluorescence (Supplementary Fig. 29). Second, no characteristic EPR signal of DMPO- $\bullet\text{OH}$ adduct can be detected for the Fe-Art M, which indicates that there is no production of $\bullet\text{OH}$ in this system (Supplementary Fig. 30). Finally, we conduct free radical quenching experiments to further identify the produced radical species during H₂O₂ activation (Fig. 4f); obviously, the signal intensity sharply decreased when benzoquinone ($\bullet\text{O}_2^-$ quencher) is added, thus confirming the production of $\bullet\text{O}_2^-$; meanwhile, no obvious signal change is noticed for $\bullet\text{OH}$ after tertiary butanol (TBA, $\bullet\text{OH}$ quencher) is added, thus indicating no generation of $\bullet\text{OH}$. For the generation of $^1\text{O}_2$,

we have compared the Fe-Art M with the pristine Art M by EPR and SOSG probe; we found that the pristine Art M framework can also generate $^1\text{O}_2$; therefore, we think that the generation of $^1\text{O}_2$ is not originated from Fe active site but from the carbon frameworks (Supplementary Fig. 31-32).

As for the HPO-like catalytic process, we first take the celestine blue (CB) as a specific HOCl/OCl⁻ probe to detect the generation ability of Fe-Art M as HPO-mimics; which shows a decrease in the peak at 648 nm and an increase in the peak at 512 nm (Supporting information 33). Therefore, after adding Fe-Art M catalyst, the obvious peak changes measured by UV indicate the production of HClO (Figs. 4g-4h and Supporting information 34). Besides, the production of HOCl/OCl⁻ for Fe-Art M was also further verified by a HOCl/OCl⁻ specific fluorescent probe, aminophenyl fluorescein (APF) (Supplementary Figure 34). In the meantime, the $\bullet\text{O}_2^-$ is also generated along with the production of HOCl (Supplementary Figure 41) since they share the same catalytic pathway for the first few steps.

In the meantime, because the substrate is H₂O₂ or H₂O₂ + NaCl, other types of ROS (like NO) are not tested. The corresponding new experiments have been added in the revised manuscript and the revised supplementary information, and shown as follows:

Fig. 4 a POD-mimetic activity via TMB-based UV-vis spectra in the presence of Art M and H₂O₂.

Fig. 4 **e** DMPO spin-trapping EPR spectra. **f** DMPO spin-trapping EPR spectra of Fe-Art M with different radical scavengers in DMSO solution. Benzoquinone (BQ, $\cdot\text{O}_2^-$ quencher), tertiary butanol (TBA, $\cdot\text{OH}$ quencher), NaN_3 ($^1\text{O}_2$ quencher). **g**, **h** Detection of HPO-mimetic activity using celestine blue (CB) as agents, the relative activity is referred to the rate of Intensity_{512 nm} / Intensity_{648 nm}.

Supplementary Figure 29. **a** Illustration of the reaction of hydroethidine (HE) by $\cdot\text{O}_2^-$. **b** Absorption spectra of HE after reaction with the catalytic produced $\cdot\text{O}_2^-$ by different samples.

Supplementary Figure 30. EPR spectra for detection of $\cdot\text{OH}$ generated from different systems in HOAc/NaOAc solution by using DMPO as trapping agent.

Supplementary Figure 31. The EPR spectra of the in situ $^1\text{O}_2$ radical detection by TEMP.

Supplementary Figure 32. a Formation of the endoperoxide upon the reaction of singlet oxygen sensor green (SOSG) with $^1\text{O}_2$. **b** Absorption spectra of SOSG after reaction with $^1\text{O}_2$ with different samples.

Supplementary Figure 33. Illustration of the oxidation of CB molecules by the catalytic produced HOCl.

Supplementary Figure 34. a Illustration of the chemical reaction of aminophenyl fluorescein (APF) by HOCl. **b** Absorption spectra of APF after reaction with HOCl produced by different samples.

Supplementary Figure 41. **a** Detection of HClO. **b** DMPO spin-trapping EPR spectra for $\bullet\text{O}_2^-$ detection.

(5) “On page 9, the authors claimed that their “Fe-Art M shows the best POD-like performance in a pH-independent manner (Fig. 4a and Supplementary Fig. 20)”. To support this claim, the authors need to use a pH-independent ROS probe, rather than TMB which works only at acidic pH, and monitor the ROS production under at least three different pH conditions, better to span the acidic, neutral, and basic ranges.”

Response to comment:

Thanks for your good comments and helpful suggestions; we are sorry that there is a mistake in the POD-like performance; it should be Fe-Art M shows a pH-dependent manner, the POD-like performance changes along with the pH values. First, we have added the pH effects on the POD-like activities; as shown in Supplementary Figure 28, we provided the pH effects on the catalytic performances of Fe-Art M as POD-mimics, and Fe-Art M displayed an optimum performance when pH was 4.

In the literature, the TMB and o-phenylenediamine (OPD) are two types of most frequently used ROS evaluation reagents; besides the TMB and OPD, we haven't found any other reagent that shows better ROS evaluation capability. But we agree with the reviewer that the TMB may work better at acidic pH. Therefore, we also take the OPD as another ROS evaluation reagent to further confirm the POD-like activity of Fe-Art M at acidic, neutral, and basic conditions. By using OPD as the detection agent, we also found that the POD-like performance of Fe-Art M at pH=4 was much better than that of pH=7 and pH 9

(Supplementary Figure 28). Therefore, we think that the OPD results are generally consistent with the TMB results. If there is any further suggestion on the ROS evaluation reagents from the reviewer, we are very happy to try with them.

Thus, we have corrected this contents on page 9 in the revised manuscript, and shown as follows: “By using the classic 3,3,5,5-tetramethylbenzidine (TMB) and **o**-phenylenediamine (OPD) colorimetric methods, the Fe-Art M shows the best POD-like performance in a **pH-dependent** manner (Figs. 4a and Supplementary Fig. 28)”.

Supplementary Figure 28. **a** Typical absorption spectra of TMB after catalytic oxidation with H₂O₂ in pH 5.6 acetate buffer. **b** Intensity at 652 nm of different samples. **c** pH-dependent absorbance changes at 652 nm by using Fe-Art M as POD-mimics. (n=3 per group, data are presented as mean ± SD). **d** Illustration of the reaction of o-phenylenediamine (OPD), **e,f** Fe-Art M show POD-like activity in a pH-dependent manner, indicating that the Fe-Art M can catalyze the oxidation of OPD to produce colorimetric reactions. Statistical significance is assessed by the Student’s two-side t-test. (*P<0.05, **P<0.01, n.s. represents no significant differences).

(6) “On page 14, the authors claimed that their hedgehog-like particle “...shows limited toxicity, especially in the absence of H₂O₂, ... (Supplementary Fig. 32)”. But, in Supplementary Fig. 32, both Fe-Art M and V-Art M caused certain ratio of cells to be stained in red (i.e., dead cells), not even to mention those that were deactivated and washed away in the sample preparation. In addition, Supplementary Fig. 32 reveals no detectable difference to the naked eyes between groups in the presence and absence of H₂O₂. In fact, to assess whether their particles are cytotoxic or not, the authors should have done cytotoxicity assays to obtain quantitative evaluation in addition to the qualitative examinations as reported in Supplementary Fig. 32.”

Response to comment:

Thanks for these important comments; we agree that the live-dead staining may not give the exact cytotoxicity results of these materials. With your help suggestion, we have performed the quantitative evaluation on cytotoxicity by using the CCK-8 assay at different conditions. From the CCK-8 tests, we found that the L929 cells exhibit high cellular viability of about 80% live ratio for the Art M in the absence of H₂O₂, while after the H₂O₂, the cellular viability ratio decreased to about 40%, indicating that the presence of H₂O₂ will lead to higher cytotoxicity (Supplementary Figure 43). It is believed that the higher cell death ratio should be attributed to the higher ROS generation capability under H₂O₂ conditions via the peroxidase-like activity. Besides, when adding H₂O₂, although the cellular toxicity increased and would kill certain amounts of cells, the cell populations could recover quickly within a few days (Supplementary Figure 43-45).

For the cellular toxicity, of course, we agree with the reviewer that some tissue cells will definitely be killed during the disinfection process, but we believe that this won't be a big problem. As long as all the bacteria are cleaned up, the tissue cells will soon recover and regenerate; however, if the bacteria are not totally eradicated, a secondary infection will happen due to the high reproduction speed of bacteria. In our disinfection process, all materials are only exposed to the surface skin; no blood vessels and deep tissues are involved. And after the bacteria are killed, the materials would be washed away immediately to reduce the potential toxicities to normal cells, blood, and tissues, and the risks of potential cellular uptake and long-term cumulative toxicity caused by endocytosis of our material are extremely low. Furthermore, we believe that the most important issue in antibacterial studies of wound disinfection should be the realization of rapid and complete sterilization, as long as the materials are stable during the disinfection

process. These new data have been added in the revised supplementary information on page 40 and shown as below:

Supplementary Figure 43. a CCK-8 assay for co-culture of bare L929 cells and Art M samples. **b** Cell viability after co-culture of bare L929 cells and Art M samples. (n=3 per group, data are presented as mean ± SD). Statistical significance is assessed by Student’s two-side t-test. (*P<0.05, **P<0.01).

Supplementary Figure 44. Representative live/dead cell staining (green: live, red: dead) for various samples after co-cultured with L929 cells.

Supplementary Figure 45. **a** CCK-8 assay for co-culture of bare L929 cells and Art M samples with or without H₂O₂. **b** Cell viability after co-culture of bare L929 cells and Art M samples with or without H₂O₂. (100 µM H₂O₂, n=3 per group, data are presented as mean ± SD). Statistical significance is assessed by Student’s two-side t-test. (*P<0.05, **P<0.01).

(7) “The authors need to quantitatively check the internalization efficiency of their micrometer-sized particles by mammalian cells.”

Response to comment:

Thank you for your helpful comment. To check whether there is an internalization phenomenon of these micrometer-sized particles by mammalian cells, we have further carried out experiments by the SEM and fluorescence microscope experiments by culturing V/Fe-Art M with L929. From the SEM image, we can see that the L929s tightly adhered onto the surfaces of V-Art M and Fe-Art M; and in general, some of the particle sizes are even larger than the cells, suggesting that there is no internalization phenomenon of these micrometer-sized particles by mammalian cells (Supplementary Figure 46). For the fluorescence microscope images of the co-cultured L929 cells and Art Ms, we found that these Art Ms are very large, which have similar sizes as that of L929 cells, and no apparent internalization phenomenon has been

noticed in Supplementary Figure 47. Furthermore, the Art Ms have also been labeled with DiI (dye with a red fluorescence), then the Art Ms are co-cultured with L929 cells, after 8 h, the Art Ms contained media are removed to avoid the interference of these suspended Art Ms. Then the L929 cells are labeled with green fluorescent dye and DAPI. The automatic inverted fluorescence microscope images of L929 cells with Art Ms show that there is almost no red fluorescence, indicating that there is almost no Art Ms cellular uptake by the L929 cells (Supplementary Figure 48).

Therefore, the risks of potential cellular uptake and long-term cumulative toxicity caused by endocytosis are extremely low. In fact, for this study, the Fe-Art M only needs a very short contact time to achieve rapid and complete sterilization of MRSA. These new results have been added in the revised supplementary information and also shown in the following section:

Supplementary Figure 46. SEM images of L929 cells after co-cultured with V-Art M and Fe-Art M, which indicates all cells adhere onto the Art Ms, but almost no cellular uptake has been observed since these hedgehog microspheres are too big for cells.

Supplementary Figure 47. Automatic inverted fluorescence microscope images of L929 cells with PBS, V-Art M, and Fe-Art M, almost no cellular apoptosis and internalization phenomenon has been observed. Some of these Art Ms are even larger than the cells which indicate that these hedgehog microspheres are difficult for cellular uptake. The co-culture time is 8 h.

Supplementary Figure 48. The Art Ms are first labeled with DiI (dye with a red fluorescence), then the Art Ms are co-cultured with L929 cells, after 8 h, the Art Ms contained media are removed to avoid the interference of these suspended Art M. Then the L929 cells are labeled with green fluorescent dye and DAPI. The automatic inverted fluorescence microscope images of L929 cells with Art Ms show that there is almost no red fluorescence, which indicates that there is almost no Art Ms cellular uptake by the L929 cells.

(8) *“The authors need to examine whether their micrometer-sized particles aggregate under protein-present conditions, to simulate the environment their particles encounter in a wound, their designed administration site.”*

Response to comment:

Thanks a lot for this instructive and highly useful comment. To further validate the concept that these Fe-Art M can be applied for wound disinfection, as suggested by the reviewer, we have further studied the particle dispersibility under protein-contained aqueous conditions. By dispersing the particle in BSA, FBS, and DMEM media, we found that the Fe-Art M particles show good aqueous dispersibility in different types of protein-contained media. It should be noted that these particles can't maintain long-term suspension in water, PBS, or in all types of protein solutions for more than 20 min since these particles are too big. However, these precipitated particles in both water or protein-contained media can be easily re-dispersed with gentle shaking for a few seconds. Therefore, we think that these particles won't have obviously aggregation problems in the protein-contained media. The corresponding new data have been added in Supplementary Figure 5-6 in the revised supplementary information and also shown below:

Supplementary Figure 5. Digital photographs of **a** original solutions, **b** Fe-Art M suspensions, and **c** Fe-C-MPs suspensions for varied durations (including 0, 5, 10, 20, and 30 min). After 5 min, the micrometer-scale Fe-C-MPs all settled, while Fe-Art M can maintain a suspension for about 20-30 min, thus showing that the dispersibility of Fe-Art M (hedgehog topography) is better than that of Fe-C-MPs (smooth surface). PBS: phosphate buffer saline; BSA: bovine serum albumin; FBS: fetal bovine serum; DMEM: Dulbecco's modified eagle medium. It is noticed that after adding Fe-Art M, the DMEM solution becomes transparent due to its porous structure induced high adsorption capability.

Supplementary Figure 6. The bottom of Fe-Art M and Fe-C-MPs solutions. The duration time is 5 min. Compared to a large amount of sediment in the Fe-C-MPs (smooth surface) group, the Fe-Art M shows almost no precipitant, thus indicating the enhanced dispersibility due to hedgehog topography.

Reviewer #3:

“In this paper, the authors synthesized a hedgehog Art M with atomic-catalytic centers to combat MRSA by mimicking the “capture and killing” process of natural macrophage integrating nanozyme-catalyzed bacterial killing via POD-like and HPO-like activities. The structure of catalytic site and catalytic mechanism are well analyzed. Such strategy is very interesting to develop artificial system for anti-infection treatment. I think that it merits the consideration for publication on Nature Communications. However, the below questions and concerns need to be addressed before acceptance.”

Response to the general comment:

Thanks for your helpful comments and suggestions to improve the quality of our manuscript. We have thoroughly and carefully corrected the manuscript based on your suggestions, and all the necessary data have been added to support our claims. Meanwhile, all the questions and concerns have been well-addressed in the revised manuscript and the revised supplementary information. Thus, we believe that the quality of this paper has been significantly enhanced. We thank you again for your great efforts.

(1) *“The authors mentioned that hedgehog artificial macrophages based on Fe-Art M can capture bacteria via its spiky topography and bioadhesive carbonaceous structure. What is the interaction between bacteria and hedgehog structure to achieve strong “capture”? Is the capture specific for MRSA or broad for any bacteria, fungi or cells?”*

Response to comment:

Thank you for the good comments, and we are sorry for the insufficient statements and control samples to demonstrate our claims clearly. In this study, in general, the strong capability to capture the bacteria of hedgehog artificial macrophages should be attributed to its spiky topography and bioadhesive carbonaceous structure. In the following section, we will carefully reply to your comments one by one.

1) For the strong interactions between bacteria and hedgehog structure, these interactions should mainly be attributed to the lipid membranes, proteins of bacteria and spiky surface of the Fe-Art M; such strong interactions have already been observed and analyzed in some earlier reports, such as a spiky surface with the cell membrane (*ACS Cent. Sci.* **2017**, 3, 839–846), hedgehog array for capture cancer cells (*Nano Lett.* **2016**, 16, 1, 766–772), spiky nanomaterials interact with and capture immune cells (*Nat.*

Nanotechnol. **2018**, 13, 1078–1086), and also spiky nanostructures with influenza virus surface (*Nano Lett.* **2020**, 20 (7), 5367-5375). These strong interactions and mechanisms between the bacteria and hedgehog structure have been well-established.

2) For the strong interactions between bacteria and carbonaceous structure, we have summarized these interactions in our earlier reviews, such as *Adv. Mater* **2017**, 29, 1602547, *Chem. Rev* **2017**, 117, 1826-1914. In brief, these interactions are derived from the abundant proteins on the bacterial surface, which can interact with the carbonaceous structure. By applying the molecular dynamic simulations, it has been depicted that the protein adsorption on carbon surface is mainly driven by the π - π stacking due to aromatic protein residues; meanwhile, it is also suggested that the hydrophobic interactions of protein structure can also promote the adsorption process. Our data in the revised manuscript also validate that a rough carbon surface will provide more intensive bacterial adhesion at the carbon interfaces, which will be discussed below.

3) To systematically prove that the strong capability of capturing bacteria by the hedgehog artificial macrophages, we have carried out some further studies via tuning the topography and roughness of microparticles. We first synthesized or obtained several different comparison samples with similar particle sizes as that of Art M, such as Fe-doped carbonous micro-particles (Fe-C-MPs, smooth surface, Supplementary Figs. 11), polystyrene microspheres (smooth surface), and SiO₂ microsphere (slightly rough surface). It can be observed that all types of Art Ms can efficiently capture abundant bacteria with no significant difference, as demonstrated by the SEM and fluorescence microscopy images (Figs. 2f-j and Supplementary Figs. 12-14). The results of Fe-C-MPs and SiO₂ microspheres indicate that both the carbonous surface and slightly rough surface exhibit limited bacterial capture capability (Supplementary Figs. 15, 16); while, the polystyrene microsphere with a smooth surface shows nearly no bacterial capture capability (Supplementary Figs. 17). Thus, these added new data can directly demonstrate the concept that these hedgehog artificial macrophages can be applied to capture the bacteria via its spiky topography and bioadhesive carbonaceous structure.

4) This capture is not specific for MRSA; the hedgehog structure shows a broad and universal capture capability to bacteria or cells. In order to provide proof that Art Ms can capture all types of bacteria, we also added the bacterial capture capability on the drug-resistant bacillus, extended-spectrum β -lactamase-producing *E. coli*. The data show that the Art Ms can also efficiently capture the bacillus, thus demonstrating its universal capture ability to bacteria (Supplementary Figure 14).

To prove that the Art Ms can capture the cells, we also have co-cultured L929 cells with Art Ms; the cells show strong interaction and adhesion with hedgehog microspheres (Supplementary Figure 46). Though this capture is not specific to certain bacteria or cells; we believe that these structures could be easily engineered into specific binding microstructures via coating or functionalizing with targeting ligands, such as mannose (*Nano Lett.* **2015**, 15, 6051–6057) or sialic acid (*Sci. Adv.* **2021**, 7, eabd3803).

What's more, we agree with the reviewer that the capture of fungi is very important and essential for further understanding the broad spectrum of capture. However, we lack fungus culture conditions and we believe that the mechanism of capture should be similar. In principle, all microorganisms could be trapped by our Art M, but the capture efficiency may vary based on their sizes. We hope that the reviewer can understand the difficulties of the fungi operation for us.

The corresponding discussion and new results have been added on page 6 in the revised manuscript and from pages 14-18 in the revised supplementary information, which are also shown as follows:

“To directly demonstrate the concept that these hedgehog artificial macrophages can be applied to capture the bacteria via its spiky topography and bioadhesive carbonaceous structure²⁹, we have incubated MRSA with different Art Ms and comparison samples (Fe-C-MPs, smooth surface, Supplementary Fig. 11), polystyrene microspheres (smooth surface), SiO₂ microsphere (slightly rough surface). It can be observed that all types of Art Ms can efficiently capture abundant bacteria with no significant difference, as demonstrated by the SEM and fluorescence microscopy images (Figs. 2f-j and Supplementary Figs. 12-14). The results of Fe-C-MPs and SiO₂ microspheres indicate that both the carbonous surface and slightly rough surface exhibits limited bacterial capture capability (Supplementary Figs. 15, 16); while, the polystyrene microsphere with smooth surface shows nearly no bacterial capture capability (Supplementary Fig. 17).”

Supplementary Figure 11. SEM images of **a** ZIF-8 large crystal, **b** carbonized ZIF-8 (C-ZIF-8), and **c** Fe-C-MPs. **d** XRD patterns of ZIF-8 and Fe-C-MPs. **e** Comparison of curve-fitted high-resolution XPS N 1s spectra of the C-ZIF-8 and Fe-C-MPs. **f** High-resolution Fe 2p XPS spectra for Fe-C-MPs. The SEM images of ZIF-8, C-ZIF-8, and Fe-C-MPs demonstrates that they share a similar size as Art Ms. As can be seen in the XRD and XPS results, the metal Fe in Fe-C-MPs exists in the form of coordination structure due to no peaks of iron oxides and the peak shift after the coordination with Fe, indicating the chemical structure of Fe-C-MPs is similar to that of Fe-Art M.

Supplementary Figure 12. SEM images of Art M, V-Art M, and Fe-Art M treated with MRSA.

Supplementary Figure 13. Fluorescence microscopy images of Control and Art M **when incubated with MRSA.**

Supplementary Figure 14. SEM images of bacterial capture capability of Art Ms when incubated with extended-spectrum β -lactamase-producing *E. coli*.

Supplementary Figure 15. SEM images of **a** MRSA and **b** extended-spectrum β -lactamase-producing *E. coli* after incubation with C-ZIF-8 and Fe-C-MPs, which indicate that the carbon material with a smooth surface shows weak bacterial capture capability.

Supplementary Figure 16. SEM images of **a** MRSA and **b** extended-spectrum β-lactamase-producing *E. coli* after incubation with SiO₂ microspheres, which indicates that the slightly rough surface exhibits relatively limited bacterial capture capability.

Supplementary Figure 17. SEM images of **a** MRSA and **b** extended-spectrum β -lactamase-producing *E. coli* after incubation with polystyrene microspheres, which indicate that the polymeric material with a smooth surface shows nearly no bacterial capture capability.

Supplementary Figure 46. SEM images of L929 cells after co-cultured with V-Art M and Fe-Art M, which indicates all cells adhere onto the Art Ms, but almost no cellular uptake has been observed since these hedgehog microspheres are too big for cells.

(2) “In Figure 2h, i, j, the authors described that “For the Fe-Art M, most of the captured MRSA are dead, while the bacteria surround the Fe-Art M all show good viability”. It is hard to understand such difference. How does Fe-Art M kill bacteria if no hydrogen peroxide is present.”

Response to comment:

Thank you for your good comment, and we are sorry for the insufficient statement. First, as demonstrated by our experiments, in the absence of H_2O_2 , the dead MRSA captured by the hedgehog Fe-Art M was caused by the spiky structure and oxidase-like catalytic ROS generation. As presented in Supplementary Fig. 18, we have analyzed the intrinsic oxidase-mimetic activity of these Fe-Art M, which suggests that the Fe-Art M exhibits very high catalytic transformation of oxygen into ROS via activation and cleavage of the O-O bond (*Sci. Adv.* **2019**, *5*, eaav5490), earlier reports have shown that the oxidase-mimetic activity-induced ROS generation can also be applied for antibacterial applications (*Nat. Commun.* **2021**, *12*, 2002). This good contact killing activity of Fe-Art M preliminarily supports the proposed concept for bacterial "capture and killing" function of hedgehog artificial macrophage.

Furthermore, compared to the time-dependent absorbance at 652 nm of Fe-Art M in air-saturated buffer, the reaction degree of Fe-Art M-catalyzed TMB oxidation show a significant increase in O_2 -saturated conditions and a sharp decrease in N_2 -saturated conditions, suggesting that the Fe-Art M can efficiently catalyze O_2 substrate into ROS (Supplementary Figure 18). However, due to the limited amount of O_2 in the bacterial growth media, the oxidase-like activity-induced ROS is not sufficient to kill all bacteria in the media. Therefore, the H_2O_2 and Cl^- are needed, which can initiate the POD-like and HPO-like activities by catalyzing the H_2O_2 and Cl^- to generate strong and high concentration ROS for efficiently eradication of MRSA (Fig. 6b).

The corresponding results have been shown in the revised manuscript and revised supplementary information:

Supplementary Figure 18. The oxidase-like activity: **a** Typical absorption spectra of TMB after catalytic oxidation in pH 4.5 acetate buffer. **b** Ultraviolet-visible (UV-vis) absorption spectra of Fe-Art M in O₂-saturated, air-saturated, and Ar-saturated sodium acetate–acetic acid buffer.

Fig. 2 g The calculated number of MRSA captured by different Art M. The inset of **g** is a magnified captured bacteria by the spikes. **h** Calculated live/dead bacteria ratio based on the fluorescence results of **i** V-Art M and **j** Fe-Art M, pH= ~5.8; data were obtained from three fluorescent microscopy images with scope 1 mm×1 mm, and presented as mean ± SD.

Fig. 6 b Real-time OD₆₀₀ values for different samples at 10 µg/mL against MRSA.

(3) “For the structure of active site in Fe-Art M, the authors concluded that the Fe-Art M have two bond distances of Fe-O/N (1.85 Å). But the evidence for Fe-O bond is not well provided.”

Response to comment:

Thank you very much for your helpful comment to further improve the quality of this manuscript. We agree with the reviewer that the evidence for the Fe-O bond should be well provided. We have tried to use the XPS O 1s spectrum to detect the Fe-O bond. However, due to the Fe element in Fe-Art M is just ~0.11 at%, the Fe-O bond could not be detected. Furthermore, the Fe-O bond could not be distinguished from Fe-N via the EXAFS fitting, since N and O are adjacent elements and have similar electronegativity and atomic radius. Therefore, in this study, we take a combination strategy of XPS data, XAS data, and structure simulation to confirm the existence of the Fe-O bond.

First, the Fe K-edge X-ray absorption near-edge structure (XANES) was tested for Fe-Art M to compare with Fe foil, Fe₂O₃, and Fe-phthalocyanine (FePc) (Fig. 3c, d and Supplementary Fig. 24), the pre-edge peak in Fe-Art M positioned at around 7114.6 eV is approximately located between FePc and Fe₂O₃, suggesting a different Fe-N coordination structure compared to the representative Fe-N₄ of FePc, the Fe-N structure display with a higher Fe valency, thus indicating the Fe atoms carry more positive charges and are partially oxidized. What’s more, EXAFS fitting results show that the coordination number of Fe-N/O is 4.9 (Supplementary Table 3), which is higher than FePc (4.0), indicating that another N/O atom is bonding with Fe. Meanwhile, the existence of Fe-Fe bond is also noticed (Supplementary Table 3), while no Fe⁰ valence is detected from the XPS spectra in Fig. 3b. Thus, we can conclude that the possible structures should be Fe₂N_{x(x=6-8)} or Fe₂N_{x(x=6-8)}O in the Fe-Art M. However, for the Fe₂N_{x(x=6-8)}, we have used the density functional theory (DFT) method to check the positive charges, it is even lower than the Fe-N₄ of FePc; meanwhile, we have fitted this kind of structure with the corresponding Fe K-edge EXAFS for Fe-Art M in R-space, the fitting curve show poor quality.

Thereafter, we think that the Fe₂N_{x(x=6-8)}O should be the most possible catalytic active center. Then, by using the DFT method again, we simulate four kinds of potential Fe₂N_{x(x=6-8)}O configurations and give

the corresponding Fe-Fe bond lengths as shown in Supplementary Figs. 26. As depicted by the atomic-resolution HAADF-STEM images in Figs. 3i, j; the distance of the Fe-Fe bond is around 2.2-2.3 Å (Fig. 3k), which is consistent with the major Fe-Fe bond (2.47 Å, Supplementary Table 3) detected from EXAFS. Only the Fe₂N₆O configuration matches well with the atomic-resolution HAADF-STEM and EXAFS data. We then further use the Fe₂N₆O configuration to fit the corresponding Fe K-edge EXAFS for Fe-Art M in R-space; the fitting curve shows good quality for this structure as shown in Fig. 3f.

Therefore, we conclude that the simulated Fe₂N₆O configuration (Fig. 3l) matches well with the results of DFT, XPS, EXAFS, and atomic-resolution HAADF-STEM images, in which the two Fe atoms are anchored by six N atoms in the graphitic framework, and the Fe-Fe are bridged by one O atom. Thus, Fe₂N₆O is the major catalytic active center in the Fe-Art M.

With the above illustration of XPS data, XAS data, and DFT calculations, we hope that it is enough to confirm the Fe₂N₆O structure and we wish that the reviewer can agree with our analysis. Furthermore, the needed data have already been added in the revised manuscript and revised supplementary information, which is also shown as follows:

Fig. 3 Fe-N_x catalytic active centers analysis of Fe-Art M. **a** Comparison of curve-fitted high-resolution XPS N 1s spectra of Art M and Fe-Art M. **b** High-resolution Fe 2p XPS spectra for Fe-Art M. **c** XANES and **d** magnified pre-edge XANES spectra of Fe foil, Fe₂O₃, FePc, and Fe-Art M. **e** Fourier transformation of Fe K-edge EXAFS of samples in R-space. **f** The corresponding Fe K-edge EXAFS fitting curve for Fe-Art M in R-space. WT-EXAFS of **g** FePc and **h** Fe-Art M. **i**, **j** Magnified HAADF-STEM image of Fe-Art M. **k** Intensity profiles obtained in positions 1 and 2, and the corresponding curves show Fe-Fe projection distances on the visual plane. **l** Simulated precise atomic coordination Fe₂N₆O structure in Fe-Art M. Atom colors: orange, Fe; gray, C; blue, N; red, O.

Supplementary Figure 24. Relation between the Fe K-edge absorption energy (E_0) and valence states for Fe-Art M and reference materials.

Supplementary Table 3. EXAFS fitting parameters at the Fe K-edge for various samples ($S_0^2=0.729$).

Sample	Shell	N^a	$R(\text{\AA})^b$	$\sigma^2(\text{\AA}^2)^c$	ΔE_0 (eV) ^d	R factor
Fe foil	Fe-Fe	8.0	2.46	0.0045	5.0	0.0017
	Fe-Fe	6.0	2.85	0.0045		
Fe ₂ O ₃	Fe-O	6.0	2.04	0.0097	0.1	0.0009
	Fe-Fe	5.9	2.97	0.0073		
	Fe-Fe	4.2	3.39	0.0073		
	Fe-Fe	5.6	3.70	0.0073		
FePc	Fe-N	4.0	1.96	0.0036	1.2	0.0015
Fe-Art M	Fe-O/N	4.9	1.85	0.0077	-1.3	0.0008
	Fe-Fe	3.2	2.47	0.0091		
	Fe-Fe	1.8	3.18	0.0091		

N^a : coordination numbers; R^b : bond distance; σ^2 : the Debye-Waller factors; ΔE_0^d : the inner potential correction. R factor: goodness of fit.

Supplementary Figure 26. Speculated four kinds of chemical structure models and the corresponding Fe-Fe bond lengths. **a, c, e, g**, top views. **b, d, f, h**, side views.

(4) “What is the optimal pH for POD-like activity and HPO-like activity, respectively? How to coordinate the two activities for bacterial killing.”

Response to comment:

Thank you for the good suggestion to improve the quality of the manuscript further. Yes, indeed, our materials show a pH-dependent manner for POD-like and HPO-like activity. As shown in Supplementary

Figure 28, we have provided pH effects on the catalytic performances of Fe-Art M as POD-mimics, and the Fe-Art M has optimum performance when pH was 4. For the HPO-like activity, we found that the activity increases when the pH value gradually decreases from pH 9 to pH 4. At pH 4, the Fe-Art M exhibits the best HPO-like performance (Supplementary Figure 35). Combining the results of POD- and HPO-like activities, we suggest that pH 4 is the optimal pH for POD-like and HPO-like activity of Fe-Art M. However, when we perform the antibacterial studies, we choose pH 6 as the experiment condition since the bacterial film exhibits such a pH value, it is more meaningful to do experiments at such a condition.

For coordinating these two activities in bacterial killing, we think we can control the experimental condition to manipulate the POD-like antibacterial activity or HPO-like antibacterial activity by controlling the NaCl concentration in lab experiments. While, it is hard to distinguish the POD- and HPO-like activities since there is always NaCl in a real bacterial infectious wound. As can be seen in our DFT calculations (Fig. 5), for the early stage of the HPO-like catalytic process, it actually takes the same catalytic pathway as that of the POD-like catalytic process. Therefore, these catalytic processes will happen together in the real bacterial disinfection applications (Supplementary Figure 41); both will make great contributions to antibacterial effects, which further explains why our Fe-Art M exhibits such good antibacterial capabilities. The corresponding data have been added in the revised supplementary information and also shown as follows:

Supplementary Figure 28. c pH-dependent absorbance changes at 652 nm by using Fe-Art M as POD-mimics. (n=3 per group, data are presented as mean \pm SD).

Supplementary Figure 35. c pH-dependent absorbance changes by using Fe-Art M as HPO-mimics, it should be noticed that the CB chromogenic reaction is not stable below pH 3 (*Nature Nanotechnology* 2012, 7, 530–535); thus we only measure the reaction starting from pH 4 to pH 9. (n=3 per group, data are presented as mean \pm SD).

POD-like activity

HPO-like activity

Fig. 5 Theoretical calculation of POD-like and HPO-like ROS catalytic reaction pathways. **a** Electron density difference image of H_2O_2 adsorption structure on $\text{Fe}_2\text{N}_6\text{O}$. Yellow contours indicate electron accumulation, and cyan contours denote electron deletion. **b** Proposed reaction path and **c** corresponding

free energy diagrams of reaction path 1 and other two potential reaction paths toward generating $\bullet\text{O}_2^-$ as POD-mimics. **d** Proposed reaction path and **e** the free energy diagrams toward generating HClO as HPO-mimics. Atom colors: orange, Fe; gray, C; blue, N; red, O; white, H; and green, Cl.

Supplementary 41. **a** Detection of HClO. **b** DMPO spin-trapping EPR spectra for $\bullet\text{O}_2^-$ detection.

(5) “For antibacterial test, 10 mM H₂O₂ is chosen in in vitro and in vivo models. How to determine 10 mM is the best concentration for these experiments. Otherwise, 10 mM is a pretty high concentration which may effectively kill bacteria or mammalian cells.”

Response to comment:

Thanks a lot for your good comment. First, we are sorry that we have made a misleading description in the supplementary information; we forget to transfer the used H₂O₂ concentration into the final H₂O₂ concentration of the system. In our study, the final H₂O₂ concentration should be 100 μM for both *in vitro* and *in vivo* experiments. The 100 μM H₂O₂ concentration is a general standard value taken by some earlier antibacterial studies, for instance, the *ACS Nano* **2016**, 10, 11000-11011. There is two reasons that we choose the 100 μM H₂O₂ concentration; one is to compare the antibacterial performances of this study with earlier researches; another reason is because this is the optimized concentration for our materials (Supplementary Fig. 49).

For cellular toxicity, of course, we agree with the reviewer that some tissue cells will definitely be killed during the disinfection process, but this won't be a serious problem for wound disinfection. As long as all the bacteria are cleaned up, the tissue cells will soon recover and regenerate; however, if the bacteria are not totally eradicated, a secondary infection will happen due to the high reproduction speed of bacteria. Therefore, we chose a relatively high H₂O₂ concentration (100 μM), which can achieve rapid and complete sterilization after a short treatment time.

The corresponding contents have been corrected in the revised manuscript and also in the revised supplementary information.

Page 5 in the revised supplementary information: “The final concentrations of materials and H₂O₂ are 10 μg·mL⁻¹ and 100 μM, respectively.”

Supplementary Figure 49. The OD₆₀₀ values and corresponding antibacterial efficiency of MRSA after treated with Fe-Art M. The concentration of H₂O₂ is 0.01, 0.05, 0.1, 0.5, 1, 5, and 10 mM, respectively. The concentration of Fe-Art M catalyst is 10 μg/mL. When H₂O₂ concentration is 0.1 mM, e.g. 100 μM, it could realize ~100% eradication towards MRSA. Therefore, the 100 μM of H₂O₂ concentration is taken for the antibacterial studies.

(6) “Based on the description in Fig. 6c, the authors concluded that the minimal inhibition concentration (MIC) for Fe-Art M+ H₂O₂ is merely 8 μg/mL. Does such MIC refer to the concentration of Fe-Art M?”

However, as mentioned in the Figure 2 by the authors, only Fe-Art M showed no significant antibacterial activity. Thus, it is questionable to use MIC to determine the antibacterial performance for Fe-Art M.”

Response to comment:

Thank you for your good comments and helpful suggestions. We agree with the reviewer that there is some problems for concluding that the MIC of Fe-Art M+ H₂O₂ is merely 8 µg/mL. The MIC should include all antibacterial ingredients or agents since H₂O₂ is also involved with the bacterial killing process. Therefore, according to the reviewer’s constructive suggestion, we have thoroughly checked and revised the corresponding section, and the corresponding statements are also shown as follows:

Page 2: “The synthesized artificial macrophage exhibits an extremely low minimal inhibition concentration (8 µg/mL Fe-Art M with H₂O₂ (100 µM)) to combat MRSA”

Page 4: “The synthesized artificial macrophage exhibits an extremely low minimal inhibition concentration (8 µg/mL Fe-Art M with H₂O₂ (100 µM)) to combat MRSA”

Page 14: “the minimal inhibition concentration (MIC) for Fe-Art M+ H₂O₂ is merely 8 µg/mL Fe-Art M with H₂O₂ (100 µM), being close to that of vancomycin (4 µg/mL) and much lower than that of 32 µg/mL V-Art M with H₂O₂ (100 µM).”

Page 16: “The synthesized Fe-Art M exhibits an extremely low minimal inhibition concentration (8 µg/mL Fe-Art M with H₂O₂ (100 µM)) to MRSA”

(7) “*Considering the biocompatibility, can Fe-Art M be degraded under physiological condition?*”

Response to comment:

Thank you for the good comment. The samples used in this study are carbonaceous materials that are very stable in physiological conditions. In our disinfection process, all materials are only exposed to the surface skin for 30 minutes; no blood vessels and deep tissues are involved; after the bacteria are killed, the materials are washed away immediately to reduce the potential toxicities to normal cells, blood, and tissues. Therefore, we believe that the biosafety of these Fe-Art M won’t be a big challenge to overcome.

For the reason why we chose the carbonaceous materials: this research focuses on verifying the concept of “design a hedgehog artificial macrophage with atomic-catalytic centers to combat MRSA by mimicking the “capture and killing” process of macrophages”; therefore, the carbonaceous material with high physiological stability is more suitable to verify this new concept. Of course, we agree with the reviewer, these materials will be difficult for internal applications, but we believe that these undegradable materials can be suitable for environmental sterilization, skin wound disinfection, and externally used bandages for burned and damaged skins;

In the meantime, considering tissue engineering, scaffold biomaterials, and other antibacterial systems that need materials to be degraded, we are now also developing and preparing hedgehog materials based on biocompatible polymers or metal oxides. We believed that, in the near future, we would also be able to realize the construction of degradable hedgehog artificial macrophages with application potentials for internal application fields that requested high biocompatibility.

We thank all reviewers again for their helpful comments and hope that this significantly revised manuscript is now acceptable for publication in *Nature Communications*.

With kind regards!

Sincerely yours,

Prof. Dr. Chong Cheng (on behalf of authors)

E-Mail: chong.cheng@fu-berlin.de or chong.cheng@scu.edu.cn

Reviewers' Comments:

Reviewer #1:

Remarks to the Author:

The authors made clear and extensive corrections to the manuscript. The points of all Referees had been addressed well. This Referee considers this MS reddest for the publication after correcting of minor typos.

Reviewer #2:

Remarks to the Author:

The authors have done additional experiments in revision. But the manuscript is still inconsistent significantly. I would recommend submission to a different journal after clarifying the following concerns:

(1) The authors claimed that their Fe-Art M and V-Art M, having rough surfaces and generating ROS, can act as artificial macrophages, to mimic macrophages' ability to capture and kill bacteria. To support their claim that the rough surfaces of Fe-Art M and V-Art M are necessary for their ability to capture bacteria, the authors reported additional SEM experiments using particles with smooth surfaces in the revised manuscript (Supplementary Figure 15-17), which revealed negligible extent of bacterial attachment on these smooth surfaces. But negligible extent of bacterial attachment on these smooth particle surfaces under SEM does not necessarily mean negligible extent of bacterial capturing by these smooth surfaces in mixtures of bacteria and particles, because the bacteria fixation step in SEM sample preparation may have removed bacteria attached on smooth particle surfaces, but not those on rough particle surfaces. To prove that the rough particle surface of Art M is acting as an actively capturing site, in situ characterization techniques rather than SEM are necessary, to avoid externally introduced interference to the bacteria-particle interactions and to exclude the possibility that particles of rough and smooth surfaces are similarly efficient in capturing bacteria but distinct in their effectiveness of retaining the attached bacteria in subsequent SEM sample preparation.

(2) This work used two Art M particles, Fe-Art M and V-Art M, but performed comprehensive characterizations only on Fe-Art M. Although V-Art M was prepared through a similar procedure as Fe-Art M, more characterizations than the currently reported ones (HRTEM, BET, HAADF-STEM, and EDX mapping) are necessary. Considering that V-Art M was claimed to be particle with atomic-catalytic centers, synchrotron XANES and EXAFS characterizations are needed to prove that V-Art M does contain atomically dispersed V centers as the authors expected.

(3) In the authors' responses to comments by reviewers, they claimed "designing highly stable hedgehog micro-sized carbonaceous particles to mimic the topography of macrophage for localized bacterial capture" as one of the two major motivations for this work. When the authors said "highly stable", they did not specify whether their use of "highly stable" was referring to chemical stability or colloidal stability. In the Supplementary Information, the authors took photos for particle dispersions in diverse media at different time-points (Supplementary Figure 5) and claimed that their Fe-Art M did not sediment appreciably at 5 min (Supplementary Figure 6). But, at only 30 min, formation of sediments in Fe-Art M dispersions was visible even to naked eyes (Supplementary Figure 5). Besides, this experiment (Supplementary Figure 5) only shows whether the particles sediment in aqueous dispersions but does not provide formation on whether the particles aggregate. Considering that Fe-Art M is a micro-sized inorganic particle, it is surprising that it did not sediment very quickly. The authors may need to characterize the density and surface chemistry of their Fe-Art M and V-Art M. By the way, materials characterizations on Fe-C-MPs (Supplementary Figure 5 and 6) are needed but absent.

(4) On page 14 of the revised manuscript, the authors claimed that Fe-Art M "shows limited toxicity especially in the absence of H₂O₂". As Fe-Art M exhibited limited antibacterial activity in the absence of H₂O₂ (Figure 2h) and were evaluated in vivo with externally added H₂O₂ (Figure 6), its cytotoxicity should be considered in the presence of H₂O₂. In the presence of H₂O₂ (100 μM), Fe-Art M reduced the viability of L929 cells to ~45% at only 2 μg/mL and continued culturing of the survived cells for another 2 days only increased the viability to ~60% (Supplementary Figure 45). In the same figure, Fe-Art M at 8 μg/mL reduced the cell viability down to ~30% and ~50% on day 1 and day 3, respectively. Such strong cytotoxicity indicates very narrow therapeutic window, because Fe-Art M at 2.5 and 5 μg/mL (in the presence of 100 μM H₂O₂) killed only 80-

85% and 90-95% bacteria, respectively, as shown in Figure 6c. Besides, it is more informative if bacterial killing data such as Figure 6c are plotted in log scale.

(5) On page 41 of the authors' response to comments by reviewers, the authors stated that "when adding H₂O₂, although the cellular toxicity increased and would kill certain amounts of cells, the cell populations could recover quickly within a few days (Supplementary Figure 43-45)".

Nevertheless, what Supplementary Figure 43-45 suggested may be contradictory to the authors' expectation. The cell viability ratios on day 3 were less than two-fold of their counterparts on day 1 (Supplementary Figure 43 and Supplementary Figure 45), with cells in the 8 µg/mL Fe-Art M group exhibiting viability ratio of only ~50% on day 3. This is surprising, considering that Art M including Fe-Art M exhibited almost no internalization by L929 cells (Supplementary Figure 47-48) and therefore the cell cultures on day 2 and 3 in CCK-8 assays may contain negligible amounts of Fe-Art M particles due to Fe-Art M removal through medium change once every day (on page 6 of the Supplementary Information file).

(6) On page 41 of the authors' response to comments by reviewers, the authors emphasized that it is acceptable for Fe-Art M to be cytotoxic because "after the bacteria are killed, the materials would be washed away immediately to reduce the potential toxicities to normal cells, blood, and tissues, and the risks of potential cellular uptake and long-term cumulative toxicity caused by endocytosis of our material are extremely low". But they did not provide experimental data to show that Fe-Art M could be washed away after the bacteria are killed, which are needed especially considering the significant attachment and spreading of L929 cells on Fe-Art M surfaces (Supplementary Figure 46).

(7) In Supplementary Figure 29 and Supplementary Figure 32, the figure legends said absorption but the labels of y-axis said PL intensity.

Reviewer #3:

Remarks to the Author:

The authors have carefully answered reviewers' questions and addressed all the concerns. In particular, the authors have added more data to support their work in the revised manuscript and supporting information. Thus I recommend it to be accepted for publication in current version.

Point-by-point response to the detailed comments by reviewers of “*Hedgehog Artificial Macrophage with Atomic-Catalytic Centers to Combat Drug-Resistant Bacteria*” with manuscript ID: NCOMMS-21-12581B.

REVIEWER COMMENTS

Reviewer #1:

“The authors made clear and extensive corrections to the manuscript. The points of all Referees had been addressed well. This Referee considers this MS reddest for the publication after correcting of minor typos.”

Response to the general comment:

Thanks a lot for your great efforts in giving us very high credits on our new findings on hedgehog artificial macrophage, and also thank you for your valuable comments to improve the quality of this manuscript. We have carefully read and thoroughly check on the typos of this manuscript; the manuscript has also been checked by a language expert carefully. And we also thank you very much for your agreement on the acceptance of this manuscript. We believe that the application of this hedgehog artificial macrophage with “capture and killing” capability and high ROS-catalytic activity will open up a promising and new pathway to develop antibacterial materials for bionic and non-antibiotic disinfection strategies.

Reviewer #2:

“The authors have done additional experiments in revision. But the manuscript is still inconsistent significantly. I would recommend submission to a different journal after clarifying the following concerns:”

Response to the general comment:

Thank you very much again for your careful reading of the manuscript and constructive comments that allowed us to improve the quality of the manuscript further. We have addressed them point by point accordingly to make the contents and conclusion more accurate and consistent, for instance, the *in situ* capture towards MRSA is directly observed by cyro-SEM and 3D reconstructions from CLSM images, the characterization of atomically dispersed vanadium metals in V-Art M, and the re-designed experiments on cellular compatibility. Please find our detailed responses to your helpful comments and kind suggestions.

Meanwhile, in our revised manuscript, supplementary information, and the response letter, all the mentioned concerns have been well-addressed and explained. All the corrections and changes in the revised manuscript and supplementary information are highlighted in red.

(1) *“The authors claimed that their Fe-Art M and V-Art M, having rough surfaces and generating ROS, can act as artificial macrophages, to mimic macrophages’ ability to capture and kill bacteria. To support their claim that the rough surfaces of Fe-Art M and V-Art M are necessary for their ability to capture bacteria, the authors reported additional SEM experiments using particles with smooth surfaces in the revised manuscript (Supplementary Figure 15-17), which revealed negligible extent of bacterial attachment on these smooth surfaces. But negligible extent of bacterial attachment on these smooth particle surfaces under SEM does not necessarily mean negligible extent of bacterial capturing by these smooth surfaces in mixtures of bacteria and particles, because the bacteria fixation step in SEM sample preparation may have removed bacteria attached on smooth particle surfaces, but not those on rough particle surfaces. To prove that the rough particle surface of Art M is acting as an actively capturing site, in situ characterization techniques rather than SEM are necessary, to avoid externally introduced interference to the bacteria-particle interactions and to exclude the possibility that particles of rough and smooth surfaces are similarly efficient in capturing bacteria but distinct in their effectiveness of retaining the attached bacteria in subsequent SEM sample preparation.”*

Response to comment:

Thanks for your very instructive comments on the bacterial capture capabilities. We agree with the reviewer that the bacteria fixation process may influence the number of bacteria attached to rough/smooth particles’ surfaces. However, since all the treatment operations towards rough/smooth particles’ surfaces are same, based on our experience data and literatures, the SEM results can usually

give the general trends of the interactions between particles and bacteria. But it is true that the shaking process during the fixation, dehydration, and centrifugation may bring a certain degree of influence on the detailed amount of captured bacteria on particle surfaces. Therefore, we agree with the reviewer that the *in situ* characterization techniques rather than SEM are necessary to avoid externally introduced interference to the bacteria-particle interactions. Currently, there are two available *in situ* characterization techniques, i.e., cryo-electron microscopy and fluorescence imaging, for directly observing the bacterial capture of rough/smooth particles.

Therefore, in order to avoid the influence of fixation, dehydration, and centrifugation steps on the attached bacteria towards rough/smooth particles' surface, we first introduce the cryo-scanning electron microscopy (cryo-SEM) technique to *in situ* observe the bacteria capturing ability on different particles, in which, the bacteria were incubated with different particles for 30 min under continuous shaking and then observed with the cryo-SEM directly, and there are no fixation, dehydration, and centrifugation steps. Since these Art M particles are micrometer scale, the cryo-TEM would be difficult for observation on the whole scale; therefore, the cryo-SEM is taken, which can give a better image of the particle-bacteria interaction in this study.

As seen in Supplementary Figures 14, 19, 21, and 23, the *in situ* cryo-SEM image shows that the Fe-Art M can efficiently capture abundant bacteria, which gives similar results to that of the SEM image in Fig. 2f. More importantly, there are only a few bacteria on the substrate, thus indicating that most of the bacteria are captured by the hedgehog particles. The results of Fe-C-MPs (Supplementary Figure 19) and SiO₂ microspheres (Supplementary Figure 21) indicate that both the carbonous surface and slightly rough surface exhibit limited bacterial capture capability, and there are a large number of bacteria on the substrates; while, the polystyrene microspheres with smooth surface show nearly no bacterial capture capability (Supplementary Figure 23). Therefore, these *in situ* cryo-SEM data have demonstrated that the rough particle surface of Art M is acting as an actively capturing site.

Furthermore, we have also added the 3D reconstructions from CLSM images of V-Art M, Fe-Art M, and Fe-C-MPs to directly observe and compare the bacterial capture ability and also the live/dead ratio on these particle surfaces (Fig. 2 and Supplementary Figure 15). Both the V-Art M and Fe-Art M can efficiently capture abundant bacteria, and there are more dead bacteria on the Fe-Art M due to its strong intrinsic oxidase-mimetic activity to produce ROS. In contrast, the CLSM image of Fe-C-MPs

microspheres indicates that the bare smooth carbonous surface exhibits limited bacterial capture capability and low bacterial killing ability (Supplementary Figure 19).

To sum up, combined with SEM, *in situ* cyro-SEM and CLSM images, we hope that the reviewer can accept our claim on Page 6 “these hedgehog artificial macrophages can be applied to capture the bacteria via its spiky topography and bioadhesive carbonaceous structure.” These data have been added in the revised supplementary information, which are also shown as below:

Supplementary Figure 14. Cryo-SEM image of Fe-Art M treated with MRSA.

Supplementary Figure 19. a Cyro-SEM image of MRSA after incubation with Fe-C-MPs. **b** 3D reconstruction from CLSM image of Fe-C-MPs when treated with MRSA. The Fe-C-MPs are marked with white lines.

Supplementary Figure 21. Cryo-SEM images of MRSA after incubation with SiO₂ microspheres.

Supplementary Figure 23. Cryo-SEM images of MRSA after incubation with polystyrene microspheres.

Supplementary Figure 15. 3D reconstructions from CLSM images of **i** V-Art M and **j** Fe-Art M when treated with MRSA. The V/Fe-Art M are marked with white lines.

(2) “This work used two Art M particles, Fe-Art M and V-Art M, but performed comprehensive characterizations only on Fe-Art M. Although V-Art M was prepared through a similar procedure as Fe-Art M, more characterizations than the currently reported ones (HRTEM, BET, HAADF-STEM, and EDX mapping) are necessary. Considering that V-Art M was claimed to be particle with atomic-catalytic centers, synchrotron XANES and EXAFS characterizations are needed to prove that V-Art M does contain atomically dispersed V centers as the authors expected.”

Response to comment:

Thanks for your important and helpful comments to improve the quality of this manuscript. We agree with you that the synchrotron XANES and EXAFS characterizations are needed to confirm the atomically dispersed V centers in V-Art M. Actually, we have tried with the synchrotron XANES and EXAFS for V-Art M when we do characterizations for the Fe-Art M. However, owing to the V content in V-Art M is only 0.10 at. %, we have tried several times by changing the testing conditions, and the results show that the V content is too low to get reliable XANES and EXAFS data. The technician for XAS measurements told us that the K-edge energy of V element (5465 eV) is much lower than that of Fe element (7112 eV);, therefore even if the Fe elements in Fe-Art M (0.11 at. %) and the V elements in V-Art M (0.10 at. %) are almost the same, it is not possible to obtain a reasonable and reliable XANES, EXAFS, and WT-EXAFS data for V-Art M. To make a better understanding on the poor data quality, we have presented the XANES spectra and the Fourier transform of V K-edge EXAFS of V-Art M and V foil in R-space, which are obvious to see that the peak is extremely illogical at the range of ≈ 1 Å in the curve for V-Art M and also there is no bond signal in WT-EXAFS.

But just as the reviewer said, we have done XRD, HRTEM, HAADF-STEM, and EDS mapping experiments and found that the V atoms are homogeneously distributed on the hedgehog particles without the formation of metal oxides or metal clusters. Besides, there is a peak shift of pyridinic N in V-Art M (398.70 eV) when compared to that in Art M (398.49 eV), indicating the formation of V-N_x/pyridinic-N (Supplementary Figure 26).(*Nano Lett.* **2020**, 20, 1252-1261)

However, we agree with the reviewer that if there is no synchrotron XANES and EXAFS data, it will be extremely difficult to solidly prove that V-Art M does contain atomically dispersed V centers. Therefore, we carried out the high-resolution HAADF-STEM images to directly observe these

atomically dispersed V centers. As shown in Supplementary Figure 34, there are also abundant uniformly dispersed paired bright atom dots on the matrix; combined with all the other data, it can be demonstrated that the V centers of V-Art M are dispersed in the form of atomic V centers. These data and the corresponding statements have been added in the revised supplementary information, which are also shown as below:

Figure. a XANES spectra and **b** Fourier transformation of V K-edge EXAFS of V-Art M and V foil in R-space. WT-EXAFS of **c** V foil and **d** V-Art M. It is to see that the peak is extremely illogical at the range of $\square 1 \text{\AA}$ in the curve for V-Art M and also there is no bond signal in WT-EXAFS.

Supplementary Figure 26. Comparison of curve-fitted high-resolution XPS N 1s spectra of Art M and V-Art M. There is a peak shift of pyridinic N in V-Art M (398.70 eV) when compared to that in Art M (398.49 eV), indicating the formation of V-N_x/pyridinic-N. (*Nano Lett.* **2020**, 20, 1252-1261)

Supplementary Figure 34. High-resolution HAADF-STEM images of V-Art M.

(3) *“In the authors’ responses to comments by reviewers, they claimed “designing highly stable hedgehog micro-sized carbonaceous particles to mimic the topography of macrophage for localized bacterial capture” as one of the two major motivations for this work. When the authors said “highly stable”, they did not specify whether their use of “highly stable” was referring to chemical stability or colloidal stability. In the Supplementary Information, the authors took photos for particle dispersions in diverse media at different time-points (Supplementary Figure 5) and claimed that their Fe-Art M did not sediment appreciably at 5 min (Supplementary Figure 6). But, at only 30 min, formation of sediments in Fe-Art M dispersions was visible even to naked eyes (Supplementary Figure 5). Besides, this experiment (Supplementary Figure 5) only shows whether the particles sediment in aqueous dispersions but does not provide formation on whether the particles aggregate. Considering that Fe-Art M is a micro-sized inorganic particle, it is surprising that it did not sediment very quickly. The authors may need to characterize the density and surface chemistry of their Fe-Art M and V-Art M. By the way, materials characterizations on Fe-C-MPs (Supplementary Figure 5 and 6) are needed but absent.”*

Response to comment:

Thanks a lot for these very helpful comments. We will carefully reply to your comments one-by-one in the following four parts:

1) For stability of the hedgehog particle, the “highly stable” refers to the chemical stability. As it is well-known that the carbonaceous materials are chemically stable in physiological conditions, which would neither change the morphology nor degrade easily, thus we believe that it is very suitable to serve as the representative hedgehog artificial macrophage with heme-mimetic catalytic sites to explore the corresponding biological activities and bacteria/particle interactions. Therefore, in this study, to verify the concept of “design a hedgehog artificial macrophage with atomic-catalytic centers to combat MRSA by mimicking the “capture and killing” process of macrophages”, we design a new spiky carbonaceous particle due to its high chemical stability. And in the process of "capturing and killing" MRSA, the material itself would not collapse or degrade. However, for many other types of materials and nanostructures, it would be difficult to ensure that they can simultaneously and stably mimic the “capture and killing” process of macrophages to combat bacteria.

2) For dispersibility and aggregation, yes, it is a very interesting phenomenon. As mentioned in our earlier response to the Reviewer one, an earlier report has already disclosed and demonstrated that the hedgehog structure can efficiently enhance the aqueous dispersibility of particles due to the decreased contact area, trapping of air, autoionization of water, etc. (Anomalous dispersions of ‘hedgehog’ particles, *Nature*, **2015**, 517, 596-599). Being consistent with the above research, we found that these Fe-Art M particles showed good aqueous dispersibility in different types of media and did not sediment appreciably at 5 min. But just as the reviewer said, the Fe-Art M is a micro-sized inorganic particle; hence it can't maintain good long-term suspension in water, PBS, or in protein solutions for more than 30 min. Furthermore, we also observe the particle aggregation of Fe-Art M and Fe-C-MPs in PBS and BSA-contained solutions by optical microscope images. Under shaking, there are no obvious aggregates. While, after the solutions were standing for 10 min, there are small aggregates for both the samples (Supplementary Figure 7), which should be caused by the π - π stacking, hydrophobic interaction, van der Waals force, etc. of the carbonaceous materials. These data and the corresponding statements have been added in the revised supplementary information, which are also shown as below:

Supplementary Figure 7. The optical microscope images of Fe-Art M and Fe-C-MPs in PBS and BSA-contained solutions after standing for varied durations. 0 min indicates that the samples are just finishing the shaking, and there are no obvious aggregates.

3) For the calculated densities of Art Ms and Fe-C-MPs, the Fe-Art M (0.783 ± 0.090 g/cm³) and V-Art M (0.685 ± 0.0180 g/cm³) are both lower than water (≈ 1 g/cm³); hence, it doesn't precipitate quickly and maintain good suspension for about 30 minutes. The Fe-C-MPs ($\approx 1.386\pm 0.111$ g/cm³) show a higher density than that of water (≈ 1 g/cm³); therefore, it will precipitate quickly. These data are provided and also shown as below:

Supplementary Table 1. The density of V/Fe-Art M and Fe-C-MPs.

Samples	V-Art M	Fe-Art M	Fe-C-MPs
ρ (g/cm ³)	0.685 ± 0.0180	0.783 ± 0.090	1.386 ± 0.111

4) For the surface chemistry and material characterizations on Fe-C-MPs, we have done XRD and XPS analysis as shown in the revised Supplementary Figure 12. As shown in the XRD data, the existing weak characteristic peaks of carbon (002) and (101) diffractions suggest that the Fe-C-MPs display low graphitic degree with abundant defects and no trace of metal-related phases. For the XPS results, compared to the C-ZIF-8, the Fe 2p spectra of Fe-C-MPs exhibits a shift of pyridinic N from 398.39 eV to 398.59 eV, indicating the formation of Fe-N_x/pyridinic-N coordination structures as that in Fe-Art M (*Adv. Mater.* **2018**, 30, 1802669). Meanwhile, the high-resolution C 1s XPS spectra for them both include C-C/C=C (284.8 eV), C-N/C-O (285.9 eV), and O=C-N/O=C-O (290.0 eV). The above XRD and XPS data jointly indicate the successful preparation of Fe-C-MPs, which shares a similar chemical structure as that of Fe-Art M. Since the Fe-C-MPs was only synthesized as a control sample with smooth surfaces to explore the bacteria/particle interaction and compare the bacterial capture capability; therefore, we didn't carry out more analysis on its precise structure, we wish that the reviewer can agree with this. These data and the corresponding statements have been added in the revised supplementary information, which are also shown as below:

Supplementary Figure 12. SEM images of **a** ZIF-8 large crystal, **b** carbonized ZIF-8 (C-ZIF-8), and **c** Fe-C-MPs. **d** High-resolution XPS spectra of C 1s in Fe-C-MPs. **e** XRD patterns of ZIF-8 and Fe-C-MPs. **f** Comparison of curve-fitted high-resolution XPS N 1s spectra of the C-ZIF-8 and Fe-C-MPs. **g** High-resolution Fe 2p XPS spectra for Fe-C-MPs. The SEM images of ZIF-8, C-ZIF-8, and Fe-C-MPs demonstrates that they share a similar size as Art Ms. As can be seen in the XRD and XPS results, the metal Fe in Fe-C-MPs exists in the form of coordination structure due to no peaks of iron oxides and the peak shift after the coordination with Fe, indicating the chemical structure of Fe-C-MPs is similar to that of Fe-Art M.

Since the comment 4 and 5 are all about the cellular toxicity of the Fe-Art M, therefore, we will make response to them together.

(4) *“On page 14 of the revised manuscript, the authors claimed that Fe-Art M “shows limited toxicity especially in the absence of H₂O₂”. As Fe-Art M exhibited limited antibacterial activity in the absence of H₂O₂ (Figure 2h) and were evaluated in vivo with externally added H₂O₂ (Figure 6), its cytotoxicity should be considered in the presence of H₂O₂. In the presence of H₂O₂ (100 μM), Fe-Art M reduced the viability of L929 cells to ~45% at only 2 μg/mL and continued culturing of the survived cells for another 2 days only increased the viability to ~60% (Supplementary Figure 45). In the same figure, Fe-Art M at 8 μg/mL reduced the cell viability down to ~30% and ~50% on day 1 and day 3, respectively. Such strong cytotoxicity indicates very narrow therapeutic window, because Fe-Art M at 2.5 and 5 μg/mL (in the presence of 100 μM H₂O₂) killed only 80-85% and 90-95% bacteria, respectively, as shown in Figure 6c. Besides, it is more informative if bacterial killing data such as Figure 6c are plotted in log scale.”*

(5) *“On page 41 of the authors’ response to comments by reviewers, the authors stated that “when adding H₂O₂, although the cellular toxicity increased and would kill certain amounts of cells, the cell populations could recover quickly within a few days (Supplementary Figure 43-45)”. Nevertheless, what Supplementary Figure 43-45 suggested may be contradictory to the authors’ expectation. The cell viability ratios on day 3 were less than two-fold of their counterparts on day 1 (Supplementary Figure 43 and Supplementary Figure 45), with cells in the 8 μg/mL Fe-Art M group exhibiting viability ratio of only ~50% on day 3. This is surprising, considering that Art M including Fe-Art M exhibited almost no internalization by L929 cells (Supplementary Figure 47-48) and therefore the cell cultures on day 2 and 3 in CCK-8 assays may contain negligible amounts of Fe-Art M particles due to Fe-Art M removal through medium change once every day (on page 6 of the Supplementary Information file).”*

Response to comments:

For questions (4) and (5) on the concerns of cellular compatibility, we will carefully reply to your comments in the following three parts:

1) For the cellular compatibility:

First, in this study, we have developed a highly efficient ROS-based non-antibiotic strategy to fight bacteria in wound disinfection. While, the ROS-based antibacterial nanoagents belong to the non-specific bactericidal materials, which can produce massive ROS to destroy bacterial cell membrane

and DNA to achieve efficient sterilization. In general, the produced ROS can't kill bacteria selectively, which will also damage the surrounding mammalian cells; therefore, it is reasonable for these Art Ms and Art Ms+H₂O₂ to show a certain degree of cytotoxicity after long-term incubation with cells, especially for the Fe-Art M with extremely efficient OXD-like and POD-like activities to produce ROS.

Second, we have carefully re-considered the added cellular compatibility data in the last revision ranged from day 1 to day 3 and different concentrations of Fe-Art M; as shown in following Figure for Response to Comments, indeed, the Fe-Art M+H₂O₂ exhibits the highest cytotoxicity with ~30% cell viability after co-culturing materials and cells for 1 day. However, we found that these cellular compatibility experiments can't match our wound disinfection treatment, where the antibacterial treatment time is only 30 min, after the disinfection process, the PBS solution was used to wash and remove the residual particles. Therefore, we believe that it is not necessary to measure the long-term cellular toxicity since the Fe-Art M will only contact the tissues or cells for 30 min during the disinfection process. Thus, we have re-designed the cellular compatibility experiments according to the actual antibacterial process during wound disinfection, that is, co-culturing cells with 10 µg/mL particles and H₂O₂ (100 µM) for 30 min, then we change the medium and wash with PBS twice to remove the particles, and then the cellular compatibility was observed immediately or after 1 day. We believe that, by this way, the results can reflect well with the actual wound disinfection application; that is, the materials quickly capture and kill the bacteria, then they could be washed away by PBS solution, and finally realizing the rapid wound healing.

Figure for Response to Comments. Cell viability after co-culture of bare L929 cells and Art M samples for continuous 1 day. (n=3 per group, data are presented as mean ± SD). Statistical significance is assessed by Student's two-side t-test. (*P<0.05, **P<0.01).

As shown in Supplementary Figure 49, when co-culturing materials and cells for a short-time of 30 min, it can maintain a cell viability of ~78%; and after removal of the particles, the cells can soon recover and reach to ~90% viability after 1 day's proliferation. Therefore, the cell viability of the Art Ms can be increased by decreasing the contacting time of materials and cells. Furthermore, it should be noted that the mammalian cells have different membrane and nuclei structures from that of bacteria, which will endow the mammalian cells different ROS sensitivity when compared with bacteria. But the structure differences and ROS sensitivity are not our research focus in this study, we will give more detailed explanations on their interesting and complicated phenomenons in our future studies. In the meantime, during the wound disinfection treatment, rapid and ~100% bacterial eradication is the first priority, and a minor degree of cytotoxicity is acceptable and always inevitable for external applications. These data and the corresponding statements have been added in the revised manuscript and supplementary information, which are also shown as below:

Supplementary Figure 49. The optical microscope of **a** adhered L929 cells when co-cultured with 10 $\mu\text{g}/\text{mL}$ Fe-Art M and H_2O_2 (100 μM) for 30 min and **b** cells after removal of Fe-Art M by changing medium and washing with PBS solution twice. It is obvious to see that there is no Fe-Art M after washing. **c** Representative live/dead cell staining (green: live, red: dead) for Fe-Art M+ H_2O_2 after co-culturing for 30 min. After then, we change the medium and use PBS to wash the particles and observe the cells after 1 day. **d** The corresponding cell viability tested by CCK-8 for Fe-Art M+ H_2O_2 after 30 min as shown in **(a)** and after 1 day as shown in **(b)**.

2) For the therapeutic window: We agree with the reviewer, in practical applications, the antibacterial agents should be sufficient, since if the bacteria are not totally eradicated, a secondary infection will happen due to the high reproduction speed of bacteria. In our re-designed cellular compatibility experiments according to the animal wound disinfection, the 10 $\mu\text{g}/\text{mL}$ Fe-Art M+ 100 μM H_2O_2 exhibits ~78% cell viability after the co-culturing time of 30 min; and after the removal of particles, the cells can soon recover and reach to ~90% viability (Supplementary Figure 49). For the concentration, at 10 $\mu\text{g}/\text{mL}$ Fe-Art M + 100 μM H_2O_2 , our materials could realize ~100% bacterial eradication. Therefore, after matching the cellular compatibility experiments with the practical wound disinfection experiments, we believe that there is no more problem with the therapeutic window.

Furthermore, we would like to note that the studies for ROS-based non-specific bactericidal materials are still in their infant stages, and this report provides a new “proof of concept” by designing the hedgehog artificial macrophage with atomic-catalytic centers for the first time, which can efficiently kill MRSA by mimicking the “capture and killing” process of macrophages. However, we agree with the reviewer’s concern on biocompatibility; we believe that the proposed new antibacterial material in this study can be mainly used for skin wound disinfection and other external sterilization applications. We have taken the skin wound disinfection as an example to illustrate their excellent bacterial eradication ability and their future application potentials for bionic and non-antibiotic disinfection.

Besides the wound disinfection, we think that these Fe-Art M can also be applied in some other areas, such as marine biofouling and face masks. In our report, the Fe-Art M can efficiently mimic HPO to generate ClO^- , which can even work at pH 8, thus making it potential for inhibiting marine biofouling (*Nat. Nanotechnol.* **2012**, *7*, 530-535). Moreover, we are also very interested in the unique cellular interactions of the hedgehog artificial macrophage; now, we are studying the possible interaction

mechanisms and trying to improve cellular biocompatibility. We hope that these hedgehog artificial macrophages and related concepts will have broad application potentials in antibacterial fields in the future.

3) **For the bacterial killing data:** we have corrected the data in log scale as suggested, which can be seen in the revised Fig. 6c and also shown as follows:

Fig. 6 c Killing ratio of MRSA treated with V-Art M and Fe-Art M at different concentrations in the presence of H₂O₂.

(6) “On page 41 of the authors’ response to comments by reviewers, the authors emphasized that it is acceptable for Fe-Art M to be cytotoxic because “after the bacteria are killed, the materials would be washed away immediately to reduce the potential toxicities to normal cells, blood, and tissues, and the risks of potential cellular uptake and long-term cumulative toxicity caused by endocytosis of our material are extremely low”. But they did not provide experimental data to show that Fe-Art M could be washed away after the bacteria are killed, which are needed especially considering the significant attachment and spreading of L929 cells on Fe-Art M surfaces (Supplementary Figure 46).”

Response to comment:

Thanks very much for your kind comments, and we are sorry for the insufficient statements here. We agree with the reviewer that additional experimental data should be provided to prove that Fe-Art M could be washed away from the tissues.

First of all, we wish to note that, for the significant attachment and spreading of L929 cells on Fe-Art M surfaces (Supplementary Figure 50), the Art Ms are co-cultured with suspension cells simultaneously to make sure there is strong interaction between them to explore the cellular uptake potential of hedgehog particles, and the experiment condition is different from the cell compatibility tests. While, in the cell compatibility experiments, the cells are first seeded for 1 day, then the suspension of hedgehog particles is added into the attached cells to observe the potential cellular toxicity.

According to the reviewer's suggestion, we have added both the *in vitro* and *in vivo* studies to observe that these materials can be washed away to reduce the potential toxicities. Here, based on our re-designed cellular compatibility experiments according to the actual antibacterial process for wound disinfection, 10 $\mu\text{g}/\text{mL}$ particles and H_2O_2 (100 μM) are co-cultured with cells for 30 min, then the medium is changed and washed with PBS twice to remove the particles, and then the adhered L929 cells was observed by optical microscope immediately. Compared to the original media (Supplementary Figure 49), it is obvious to see that there is no Fe-Art M particles after washing, thus suggesting that the particles could be totally washed away *in vitro*.

For the actual *in vivo* wound bacterial disinfection experiments, we apply the rabbits' infected wound model to study the removal capability of the Fe-Art M particles. The treatment process is the same as the wound disinfection experiments; in brief, we treat the infected wound and bare wound with our materials for 30 minutes, respectively, then we remove the materials and wash the wounds with PBS solution twice. Finally, we cut the treated wound tissue to do *in situ* cyro-SEM without any fixation and dehydration steps. And the results show that no Fe-Art M particles can be found on the wound tissue surface (Supplementary Figure 57), which could verify that our materials could be washed away after the antibacterial treatments. These data and the corresponding statements have been added in the revised manuscript and also in the revised supplementary information, which are also shown as below:

Page 18 in the revised manuscript: "For living/dead fluorescence staining, we first co-culture cells with 10 $\mu\text{g}/\text{mL}$ particles and H_2O_2 (100 μM) for 30 min, then change the medium and wash with PBS twice to remove the particles, and then observe live/dead cells immediately or after 1 day."

Page 18 in the revised manuscript: “The antibacterial treatment time is 30 min; after the disinfection process, the PBS solution was used to wash and remove the residual samples.”

Supplementary Figure 49. The optical microscope of **a** adhered L929 cells when co-cultured with 10 $\mu\text{g}/\text{mL}$ Fe-Art M and H_2O_2 (100 μM) for 30 min and **b** cells after removal of Fe-Art M by changing medium and washing with PBS solution twice. It is obvious to see that there is no Fe-Art M after washing.

Supplementary Figure 57. The digital photographs of the **a-d** infected wound and **e** cryo-SEM images, **f-i** bare wound and **j** cryo-SEM images. It is obvious to see that there are no Fe-Art M particles on the wound, only a few of dead bacteria can be found on the wound after disinfection.

(7) *“In Supplementary Figure 29 and Supplementary Figure 32, the figure legends said absorption but the labels of y-axis said PL intensity.”*

Response to comment:

Thank you very much for pointing out this mistake. We have corrected the figure legends in the revised supplementary information, which is expressed as “**The fluorescence spectra of**” on page 27 in Supplementary Figure 36, on page 30 in Supplementary Figure 39, and on page 32 in Supplementary Figure 41.

Finally, thank you very much for your professional comments and constructive suggestions, which are greatly appreciated and highly helpful for further improving the manuscript’s quality. We sincerely hope that the significantly revised manuscript and supplementary information have addressed all your concerns and suggestions.

Reviewer #3:

“The authors have carefully answered reviewers' questions and addressed all the concerns. In particular, the authors have added more data to support their work in the revised manuscript and supporting information. Thus I recommend it to be accepted for publication in current version.”

Response to general comment:

Thanks a lot for your great efforts in giving us many valuable comments to improve the quality of this manuscript. And we also thank you very much for your agreement on the acceptance of this manuscript. We believe that the application of this hedgehog artificial macrophage with “capture and killing” capability and high ROS-catalytic activity will open up a promising and new pathway to develop antibacterial materials for bionic and non-antibiotic disinfection strategies.

We thank all reviewers again for their helpful comments and hope that this significantly revised manuscript is now acceptable for publication in *Nature Communications*.

With kind regards!

Sincerely yours,

Prof. Dr. Chong Cheng (on behalf of authors)

Reviewers' Comments:

Reviewer #2:

Remarks to the Author:

The authors have added extensive experiments to address most concerns to the last version manuscript and provided explanations to those which they could not address. I would recommend acceptance for publication.